



# The impact of meteorological analysis uncertainties on the spatial scales resolvable in $CO_2$ model simulations

Saroja M. Polavarapu[1], Michael Neish[1], Monique Tanguay[2], Claude Girard[2], Jean de Grandpré[3], Kirill Semeniuk[3], Sylvie Gravel[3], Shuzhan Ren[4], Sébastien Roche[5], Douglas Chan[1] and Kimberly Strong[5]

[1]Climate Research Division, Environment and Climate Change Canada, Toronto, Ontario, M3H 5T4, Canada
[2]Meteorological Research Division, Environment and Climate Change Canada, Dorval, Québec, H9P 1J3, Canada
[3]Air Quality Research Division, Environment and Climate Change Canada, Dorval, Québec, H9P 1J3, Canada
[4]Air Quality Research Division, Environment and Climate Change Canada, Toronto, Ontario, M3H 5T4, Canada
[5]Dept. of Physics, University of Toronto, Toronto, Ontario, Canada

*Correspondence to*: Saroja Polavarapu (Saroja.polavarapu@canada.ca)

**Abstract.** A new model for greenhouse gas transport has been developed based on Environment and Climate Change Canada's operational weather and environmental prediction models. When provided with realistic posterior fluxes for $CO_2$, the $CO_2$ simulations compare well to NOAA's CarbonTracker fields, and to near surface continuous measurements, columns from the Total Carbon Column Observing Network (TCCON), and NOAA aircraft profiles. This coupled meteorological and tracer transport model is used to study the atmospheric modulation of $CO_2$ transport. The predictability of $CO_2$ due to initial state sensitivity is shorter than that for the temperature field but is consistent with the predictability of the wind fields. However, when broken down into spatial scales, $CO_2$ has predictability at the very largest scales due to long time scale memory in surface $CO_2$ fluxes as well as in land and ocean surface forcing of meteorological fields. The predictability due to the land and ocean surface is most evident in boreal summer when biospheric uptake produces large spatial gradients in the $CO_2$ field. Predictability errors provide an upper limit for errors arising solely from the use of uncertain meteorological analyses. When considering meteorological analysis errors, $CO_2$ can be defined only on large scales. Thus, there is a spatial scale below which information cannot be obtained simply due to the fact that meteorological analyses are imperfect. Compared to the spatial scales resolvable in the context of imperfect atmospheric analyses, the differences between two sets of posterior fluxes are resolvable only for very large scales. Similarly, the impact of convective tracer transport exceeds that due to atmospheric analysis errors for only the largest spatial scales.

## 1 Introduction

Atmospheric observations of $CO_2$ are important for understanding the global carbon cycle and its response to perturbations from anthropogenic emissions into the atmosphere. The global carbon budget is routinely updated using atmospheric and ocean measurements in conjunction with a careful accounting of anthropogenic emissions (LeQuéré et al., 2015). Since the greatest uncertainty in the quantified exchanges of carbon between the atmosphere, ocean and land reservoirs is associated





with the terrestrial biosphere flux (Friedlingstein et al., 2014), this component is determined as the residual between anthropogenic emissions and atmospheric and ocean changes (which are relatively well constrained by measurements). The growth in global atmospheric $CO_2$ is attributed to anthropogenic emissions but superimposed on this trend are seasonal and interannual variations which are largely attributed to the terrestrial biosphere (Ciais et al., 2013). Specifically, observations

of the 13C isotope (which is fractionated during photosynthesis by land plants so its relative concentration is indicative of the terrestrial biosphere) reveal that the interannual and seasonal variations of the global atmospheric $CO_2$ budget are due to the terrestrial biosphere (Keeling, 2005). At the same time, biospheric uptake is influenced by climate variations (Nemani et al., 2003; Friedlingstein et al., 2006; Zhao et al., 2011). Specifically, the El Niño Southern Oscillation (ENSO) signal and volcanic eruptions can explain 75% of the variations in the terrestrial biospheric uptake (Raupach et al., 2008). Thus,

atmospheric observations are important for determining the global terrestrial biospheric flux, while temporal biospheric flux variations explain atmospheric variability of $CO_2$ concentrations on seasonal and interannual timescales.

Beyond the global budget, an important challenge is to understand how temporal variations in atmospheric sources and sinks reflect the interplay between natural processes and anthropogenic perturbations and also the feedback between the carbon cycle and climate variations. By combining atmospheric observations with atmospheric models, a spatial distribution

of fluxes of $CO_2$ between the Earth's surface and the atmosphere can be obtained through inverse modelling (e.g. Rödenbeck et al., 2003; Patra et al., 2005; Baker et al., 2006a; Peylin et al., 2013; Chevallier et al., 2014). Such "flux inversions" performed on the global domain and incorporating only around 100 or so $CO_2$ observation stations near the surface are able to constrain the global atmospheric carbon budget, capture interannual and seasonal variations and attribute these to the terrestrial biosphere (Rödenbeck et al., 2003; Peylin et al., 2013). In fact, only a few observation sites representative of

background values (far from $CO_2$ sources and sinks) are needed to constrain the global $CO_2$ budget because variations in the vicinity of source/sink regions are smoothed out by the time these locations are reached (Bruhwiler et al., 2011; Keeling, 2005). However, with more surface observations, the retrieved flux uncertainties can be reduced and the ability to retrieve smaller spatial scales is improved (Bruhwiler et al., 2011). Moreover, with the desire to understand the interplay between the natural and anthropogenically perturbed processes, observations near source and sink regions become important. Such

observations will be influenced by atmospheric variations on diurnal, synoptic, seasonal and interannual time scales. On the diurnal timescale, the variation of $CO_2$ due to uptake by plants through photosynthesis in sunlit hours is strongly modulated by turbulent transport through the planetary boundary layer (PBL) which also evolves throughout the day. This is the so-called rectifier effect which helps to explain the annual mean north-south gradient of CO2 (Denning, 1995). Specifically, the uptake of $CO_2$ by plants during the spring growing season occurs when the PBL is generally unstable and deeper while in

winter $CO_2$ increases due to biospheric respiration can build up when the boundary layer is stable and shallow. In addition, poleward heat transport by baroclinic disturbances is stronger in winter preferentially transporting high $CO_2$ values north relative to the summer, in the northern extratropics (Chan et al., 2008). Synoptic scale systems also influence $CO_2$ evolution, particularly in northern midlatitudes where advection can explain up to 70% of day-to-day variability (Parazoo et al., 2008). On the interannual time scale, variations in biospheric uptake can be partially attributed to climate variations



(Patra et al., 2005) and flux inversion systems are able to attribute interannual variability of the global $CO_2$ budget to the tropical biosphere (Baker et al., 2006a).  Because atmospheric observations of $CO_2$ contain the signals of both surface fluxes and atmospheric variations, and they are needed for data assimilation in state estimations or flux inversions, it is important to be able to accurately characterize and model how the atmosphere modulates $CO_2$ evolution.

5        In a flux inversion system, observations of $CO_2$ concentration are used to solve for surface fluxes.  To relate the surface flux to an atmospheric concentration, an offline atmospheric transport model is typically used.  In doing so, the mismatch between observed and modelled concentrations can be inverted to estimate a flux increment, but the atmospheric model's transport is assumed to be perfect (Baker et al., 2006b).  The fact that it is not, means that one source of uncertainty in the inverse problem is not accounted for.  For this reason, the need to characterize "transport errors" has been well

recognized and has led to the formation in 1993 of an international group called TransCom (http://transcom.project.asu.edu/) focused on understanding and quantifying the contribution of transport errors to flux estimates from  inverse models.  Because different models are used by different flux inversions, retrieved fluxes based on multiple inversions may be more reliable than those from a given system since the unknown transport error (if it is random) may be averaged out.  Moreover, the sensitivity of flux inversion results to transport error can be identified since different models and hence different

transport errors are used.  Transport error generally refers to the deviation of $CO_2$ prediction from an (unknowable) true value due to the use of an imperfect transport model and thus includes errors due to model formulation (e.g. convective or boundary layer parameterizations or advection schemes), the use of imperfect meteorological analyses, and uncertain $CO_2$ initial conditions.   Transport errors have been found to be an important source of errors in flux inversions (e.g. Chevallier et al., 2014; Chevallier et al., 2010; Houweling et al., 2010; Law et al., 1996).  The sources of transport error arising from

model formulation include errors in the representation of:  mixing in the planetary boundary layer (Denning et al., 1995), vertical mixing in the free atmosphere (Stephens et al., 2007, Yang et al., 2007), synoptic scale and frontal motions (Parazoo et al., 2008) and convective transport (Ott et al., 2011; Parazoo et al., 2008).

        The uncertainty in meteorological analyses is also an important source of transport error (Liu et al., 2011).   In an effort to address this type of uncertainty which cannot be accounted for in flux inversion systems, NOAA's CarbonTracker

attempted to perform an ensemble of inversions using different atmospheric analyses as well as different prior flux sets1. With a coupled meteorological and tracer forecast model (as used by Liu et al., 2011), the impact of meteorological uncertainties on tracer transport are more easily identified.  Recently, coupled meteorological/tracer forecast models have been developed at operational centers such as the European Centre for Medium Range Weather Forecasting (ECMWF) (Agusti-Panareda et al., 2014) and NASA Goddard's Global Modelling and Assimilation Office (GMAO) (Ott et al., 2015).

In these operational systems, short-term predictions of greenhouse gases are produced and satellite observations are assimilated at ECMWF (Massart et al., 2016).  Such products may provide useful background or a priori information for

---

[1] This approach was later abandoned because only one of the two sources of atmospheric analyses provided the necessary convective mass fluxes needed for accurate tracer transport.  See
http://www.esrl.noaa.gov/gmd/ccgg/carbontracker/CT2013B_doc.php#tth_sEc7.2





satellite retrievals as well as providing boundary conditions for regional flux inversions. Coupled meteorological and $CO_2$ data assimilation systems also provide useful information on the reliability of correlations between meteorological and $CO_2$ transport errors (Kang et al., 2011) and on the temporal propagation of the observed signal in the context of transport errors (Kang et al., 2012).

5       The goal of this work is to better understand how atmospheric transport is modulating the $CO_2$ mixing ratio and how it is impacted by uncertain meteorological analyses. While Liu et al. (2011) have shown that meteorological forecast errors in the presence of $CO_2$ gradients produce $CO_2$ transport errors, here we consider, for the first time, the predictability of $CO_2$ and the spatial scales identifiable in the context of imperfect meteorological analyses. Using a coupled meteorological and tracer transport model, we first study $CO_2$ predictability on weather (2 weeks) and seasonal time scales in order to obtain

10   an upper limit to forecast errors arising from initial state errors. Then the spatial scales of errors in $CO_2$ arising from the use of imperfect atmospheric analyses are determined. It is shown that there is a spatial scale below which $CO_2$ cannot be retrieved, simply due to the presence of meteorological analysis errors. The spatial scales resolved from using different posterior flux estimates and the spatial scales of model errors (such as the lack of convective transport of tracers) are then compared to the scales retrievable in the context of imperfect atmospheric analyses.

15       The article is organized as follows. In section 2, we describe a new model for greenhouse gas transport based on an operational weather and environmental prediction model. In section 3, this model is assessed in terms of its meteorology (24 h forecasts are compared to reanalyses used in flux inversions) and its $CO_2$ transport. The atmospheric modulation of $CO_2$ transport is presented in section 4, starting with the predictability problem on weather timescales and proceeding to the issue of predictability on seasonal timescales. The focus of the analysis is on the spatial scales of $CO_2$ transport errors due to

20   uncertain initial conditions and uncertain meteorological analyses. Results are discussed and summarized in Section 5.

## 2 Model Description

In order to simulate greenhouse gas evolution, we can take advantage of the comprehensive forecast models already developed for operational environmental prediction at Environment and Climate Change Canada (ECCC). However, it was necessary to adapt these models for the purpose of tracer transport on multi-annual time scales. Specifically, it was

25   necessary to implement a mass conservation scheme, redesign the tracer variables, modify the vertical mixing in the boundary layer, and add convective tracer transport. The basic modelling tools are described in the next subsection while the adaptations needed for this work are presented in subsection 2.2 (mass conservation and tracer variable definitions), 2.3 (horizontal diffusion), 2.4 (convective transport) and 2.5 (boundary layer mixing). The coupled meteorology and tracer transport forecasting system is described in subsection 2.6.



## 2.1 The Canadian operational environmental prediction models

For many decades, the Canadian Meteorological Centre (CMC) has been producing operational weather forecasts for public dissemination. Since 24 February 1997, these forecasts have utilized the Global Environmental Multiscale (GEM) model (Côté et al., 1998a,b; Girard et al., 2014). GEM is a gridpoint model which solves the hydrostatic (global domain) or
nonhydrostatic (Yeh et al., 2002) (regional domain) primitive equations using a hybrid terrain-following vertical coordinate (Girard et al., 2014). As of Feb. 2013, the grid spacing of the global model is roughly 25 km, originally using a regular lat-lon grid, and since Dec. 2015, a Yin-Yang grid (Qaddouri and Lee, 2011). There are 80 vertical levels spanning the surface to 0.1 hPa. The usual physical processes of radiation (Li and Barker, 2005), boundary layer mixing (Bélair et al., 1999), shallow and deep convection (Kain and Fritsch, 1990; Kain, 2004), orographic gravity wave drag (McFarlane, 1987), and
nonorographic gravity wave drag (Hines, 1997a,b) are included in all model configurations. The land surface model and assimilation scheme are described in Bélair et al. (2003a,b). More details of the physics package are found in Mailhot et al. (1998).

Operational air quality forecasts have been produced by CMC since 2001 in order to provide real time forecasts of the air quality health index on a limited area domain covering most of North America. As of 18 November 2009, these
forecasts have utilized GEM-MACH (Modelling Air quality and CHemistry) (Moran et al., 2010; Robichaud and Ménard, 2014; Makar et al., 2015). GEM-MACH is a version of GEM in which complete tropospheric chemistry (involving over 100 chemical reactions) is modelled online, where "online" refers to the fact that the chemistry module is fully integrated into the meteorological model time step. The operational products involve an analysis of ground-level ozone, fine and coarse particulate matter (PM2.5 and PM10), NO2 and SO2 over a limited area domain covering North America (Robichaud and
Ménard, 2014). The grid spacing currently used is 10 km horizontally, with 80 vertical levels from the surface to 0.1 hPa. The operational forecasts are driven by time evolving meteorological boundary conditions from the operational regional deterministic prediction system (Fillion et al., 2010, Caron et al., 2015), while the chemical boundary conditions are defined using predetermined seasonally-averaged states. A global version of GEM-MACH is also in development for the purpose of providing boundary conditions for the regional model and a parameterized stratospheric chemistry model (McLinden et al.,
2000) is used for UV index forecasting.

Our primary interest is in global greenhouse gas distributions, thus we chose to use the global GEM-MACH configuration. In the future, extensions of our system for regional greenhouse gas simulations could be based on the operational regional GEM-MACH configuration. By definition, an operational forecasting system is constantly changing. Since our period of interest commences in 2009 with the launch of GOSAT on 23 January and since the upper boundary of
GEM was raised from 10 to 0.1 hPa on 22 June 2009 (Charron et al., 2012), this stratospheric model configuration was chosen. However, for computational expediency (especially during the model development phase) the grid spacing was coarsened to 0.9° (400×200 grid points) which is roughly twice the grid spacing used by CMC's global deterministic prediction system (800×600 grid points) in 2009. The model time step is 15 minutes.





Our primary focus in this work is global carbon dioxide simulation, but extensions for other greenhouse gases (methane and CO) are also being developed. For that we employ a simple parameterized climate chemistry involving a single OH reaction for the methane and CO simulations. This chemical module is activated from the GEM-MACH chemical interface which is also used to handle emission inputs. This simplified model version which only includes the treatment of

greenhouse gas (chemistry and transport) will be called GEM-MACH-GHG hereafter.

### 2.2 Mass conservation and tracer variable definitions

The dry air mass of the atmosphere is known to be constant, since changes in trace gases are very small (Trenberth 1981, Trenberth and Smith, 2005). However, as with many weather and climate prediction models, GEM does not conserve dry air mass. Specifically, GEM loses 0.1 hPa in global mean surface pressure during a 10-day forecast, which is precisely the same

rate as that seen in the ECMWF model (Diamantakis and Flemming, 2014). This loss is only 0.01% of the global mass and thus there is a negligible impact on medium-range weather forecasts. However, for longer simulations relevant for climate timescales, this error can accumulate. Thus GEM (as does the ECMWF and other models) has a parameter to allow the global mean surface pressure to be conserved by adding a spatially uniform adjustment to each grid cell. The constant is determined by the constraint that the air mass at the end of the dynamics step equal that at the start of the step. For our

simulations of tracer transport, it is necessary to use this switch to enforce the conservation of dry air mass.

The advection scheme in GEM uses a semi-Lagrangian approach as it affords longer time steps and the computation efficiency desirable in an operational context. However, semi-Lagrangian schemes are well known to be non-conservative (Williamson, 1990; Staniforth and Côté, 1991). GEM is a grid point model, and the semi-Lagrangian approach involves first determining the upstream positions of the grid cells (using a 3-dimensional trajectory in our case) and then interpolating the

value of the advected field at these locations. During this first step, gradients in the wind field will limit the accuracy of advection and during the second step, errors associated with the interpolation of the advected field can deteriorate species mass conservation. By combining a finite volume approach with the semi-Lagrangian scheme, it is possible to devise inherently mass-conserving schemes such as SLICE (Semi-Lagrangian Inherently Conserving and Efficient) (Zerroukat and Allen, 2012). Indeed, this scheme is being investigated for implementation with GEM in the future. In the meantime, an

interim solution is needed. The approach taken here, as in Diamantakis and Flemming (2014), is to adopt a global mass fixer. A global mass fixer scheme computes the global mass of tracer at the beginning and end of the advection step and then distributes the change in global mass among various grid cells. There is no unique way of distributing the mass spatially, so different choices result in different mass fixer schemes. While a few approaches were tried and tested with a few different chemical species (ozone, $CH_4$ and $CO_2$), the one which produced the most physically desirable results was that

due to Bermejo and Conde (2002). In this scheme, the global mass is distributed according to the smoothness of the field. That is, mass is preferentially adjusted where gradients are larger (and interpolation error is known to be larger). The exact scheme used in GEM is precisely that selected by ECMWF and it is described in Diamantakis and Flemming (2014, section





3.1). Note that for greenhouse gas transport, the ECMWF forecast model uses a proportional mass fixer instead of the Bermejo-Conde scheme (Agusti-Panareda et al., 2014).

For tracers, another issue with the interpolation step of the semi-Lagrangian advection scheme is the potential creation of spurious subgrid scale structure due to Gibbs effects. To avoid this problem, GEM uses a monotonic interpolation scheme (Bermejo and Staniforth, 1992) which combines high-order and low-order interpolation schemes to prevent overshoots and undershoots, thereby preventing the formation of spurious extrema. However, just as Flemming and Huijnen (2011) noted, enforcing monotonicity was found to worsen the problem of mass non-conservation because it will tend to diffuse gradients. Therefore, monotonicity was replaced by an iterative locally mass conserving (ILMC) scheme (Sorensen et al., 2013). The idea behind the ILMC scheme is to locally preserve the shape of the field and its gradients by distributing the excess (or deficit of) mass due to spurious extrema in the cubic interpolated field to ever increasing shells around the upstream departure point but in such a way that spurious extrema are avoided. By design, the use of ILMC does not impact our lack of global mass conservation, so it is used for preserving positive definiteness of tracer fields in place of a quasi-monotonic interpolation scheme. The ILMC is performed first, and then Bermejo and Conde (2002) scheme is applied to the ILMC-corrected field. Details on the implementation of monotonic and mass-conservation schemes in GEM and their impact on species transport are found in de Grandpré et al. (2016).

The primitive equations solved by GEM are naturally written in terms of a moist density or pressure variable. However, when tracers are defined as mixing ratios with respect to moist air, tracers become coupled to the water vapour evolution. Thus one of two options must be taken. Either the tracer variable must be redefined every time the water vapour field is modified in the model, or the tracers must be defined as mixing ratios with respect to dry air (e.g. Neale et al., 2010). The latter option has the advantage that measurements of $CO_2$ are frequently made in terms of dry mole fraction and are thus more easily related to a mixing ratio with respect to dry air. Since the nominal tracer equation within GEM assumed a moist mixing ratio (MMR), for this work, GEM was modified to permit tracers to be defined as mixing ratios with respect to dry air (DMR). This involved: (1) modifying the global mean surface pressure adjustment to ensure global dry air mass conservation, (2) converting the mass fixer scheme (i.e. the Bermejo-Conde scheme) and the ILMC to deal with DMR, (3) modifying the vertical diffusion equation to handle DMR, and (4) ensuring emissions are correctly inserted into GEM's bottom model layer. Once all of these changes had been made, the tracer mass change in a given time step was found to still display a coupling to the water vapour change in that same time step. The reason was found to be due to the fact that the continuity equation in GEM does not account for the change in surface pressure due to the change in water vapour which occurs as a result of mass flux at the Earth's surface (i.e. because of precipitation or evaporation). By adding a new adjustment to surface pressure field after the physics step to account for the change in global water vapour mass, and correspondingly redefining the tracer variable using a vertical regridding approach similar to Jöckel et al. (2001) except for the use of dry pressure coordinates, global mass conservation of $CO_2$ was obtained during a model forecast. This deficiency in GEM's continuity equation is not unusual as it was also present in NASA GMAO's GEOS5 model (Takacs et al., 2015). The adjustment to the model dynamics done here to account for the modified continuity equation is rather similar to that



described in section 2a of Takacs et al. (2015) although our work was done independently. The main difference is that only water vapour mass is considered here whereas cloud liquid water and cloud ice mass are also considered in Takacs et al. (2015) (although the authors note that additional masses due to liquid and ice phases are negligible). As these authors demonstrate, without the correction to the model's continuity equation, dry air mass conservation is not assured. They also

note that in the context of an assimilation cycle, further constraints on dry air mass are needed during the analysis and initialization steps. Furthermore, they show that the impact of the errors is evident in the mismatch of global water vapour tendency based on the water vapour field itself versus that based on precipitation minus evaporation changes for reanalysis products such as MERRA (Rienecker et al., 2011) and ERA-Interim (Dee et al., 2011).

**2.3 Horizontal diffusion**

It is standard to apply an explicit high order (typically $\nabla^6$) diffusion operator to the meteorological fields in applications involving the global GEM model. It is arguably consistent therefore to apply the same level of diffusion to the tracer fields. However, because the constituent variables are defined as mixing ratios, the operator is not mass conservative. Rather it will conserve mixing ratio. Because of the effort involved in obtaining a careful accounting of tracer mass (described in section 2.2), it was decided to not apply any horizontal diffusion to the tracers.

**2.4 Convective Transport**

The parameterization of deep convection most frequently used by GEM and GEM-MACH is due to Kain and Fritsch (1990). The Kain and Fritsch (KF) scheme is based on a single column bulk mass flux approach. Entrainment of ambient air into the cloud environment associated with updrafts and downdrafts is proportional to the corresponding mass flux and inversely proportional to the cloud radius. So narrower convective towers experience more entrainment of lower-buoyancy ambient

air and consequently have less intensity and lower cloud tops. The KF scheme is used in several forecast models (e.g. Japanese Meteorological Agency model, Saito, 2012; Bologna Limited-Area Model (BOLAM), Lagouvardos et al., 2003; High-Resolution Limited-Area Model (HIRLAM), Eerola, 2013; Weather Research and Forecasting Nonhydrostatic Mesoscale Model (WRF-NMM), Gallus Jr. and Bresch, 2006). With the Kain-Fritsch scheme, GEM has a good representation of convectively coupled waves and is able to capture the Madden-Julian oscillation (Lin et al., 2008).

25       The version of the Kain-Fritsch scheme used for global deterministic prediction, which was our starting point, had to be modified for the purpose of greenhouse gas transport. The original parameters resulted in too frequent penetration of convection into the stratosphere in the tropics. This is because the updraft core radius had been set to 1500 km globally. This is a reasonable value for extratropical convection over land, and since the target region of ECCC's forecasting systems is Canada, it is a valid choice. However, observations ( Lucas et al., 1994 and references therein) show that the updraft core

radius varies with latitude and from land to ocean. Thus over the oceans, a value of 900 km is used in the tropics with 1000 km in the extratropics. Over tropical land grid points, a value of 1200 km is used. With these settings, tropical convection overshoots fall in the correct range (below 80 hPa). However, the model climatology is altered. The parameter changes





impact land-sea contrasts which influence stationary Rossby wave generation and the spin-up of the Brewer-Dobson circulation in the stratosphere. They also affect flow over topography which impacts orographic wave drag and its associated transport circulations. An animation (**Fig. S1**) shows a comparison of the impact of changing Kain-Fritsch parameters versus that of adding convective tracer transport on $CO_2$ evolution. As expected, the change in parameters

impacts the stratospheric distribution after a few months of simulation, whereas the introduction of tracer transport through deep convection impacts the tropical tropospheric $CO_2$ distribution at all times as well as the northern hemisphere in summer. The zonal mean values are small but so too are zonal standard deviations (both are less than 0.3 ppm). Another animation which compares the impact of changing KF parameters versus that of adding convective tracer transport on column mean or $XCO_2$ (**Fig. S2**) reveals that the magnitude of the impact is smaller for the change in parameters (maxima of

0.3 ppm) than for the introduction of tracer transport through deep convection (maxima of 0.8 ppm).

## 2.5 Vertical mixing

Turbulence in the Planetary Boundary Layer (PBL) is important for the transport of heat, momentum, moisture and constituent fluxes from the surface to the atmosphere. The PBL scheme in GEM is described in Bélair et al. (1999), Benoit et al. (1989) and Mailhot and Benoit (1982). Since a summary of GEM's PBL scheme was recently presented in McTaggart-

Cowan and Zadra (2015) and Aliabadi et al. (2016), we note here only the main differences between GEM's PBL parameterization and those of other models used for greenhouse transport or flux inversion.

The vertical transport of greenhouse gas fluxes follows that of other constituents and is described by:

$$\frac{\partial C}{\partial t} = -\frac{1}{\rho}\frac{\partial}{\partial z}\left(\rho\overline{w'C}\right) = \frac{1}{\rho}\frac{\partial}{\partial z}\left[\rho K\left(\frac{\partial C}{\partial z} - \gamma\right)\right] \tag{1}$$

where $C$ is the constituent mixing ratio on resolved scales, $\rho$ is air density, $w'$ is subgrid scale vertical velocity and the

product with an overbar is the vertical flux of constituent due to subgrid scale turbulence. The first equality denotes the impact of subgrid scale motions on the resolved constituent distribution. The subgrid scale flux is parameterized through the second equality, and is a function of the vertical gradient of the constituent. $\gamma$ is a counter gradient flux relevant for an unstable PBL. $K$ is the thermal eddy diffusivity and it depends on properties of the flow. Most models define $K$ diagnostically in terms of an eddy length scale and local gradients of wind and virtual potential temperature. Such schemes

are said to invoke a first-order closure. However, GEM's scheme specifies $K$ through a prognostic equation for turbulent kinetic energy and is thus deemed a 1.5 order closure scheme (Holtslag, 2015). When $K$ is determined diagnostically through local flow properties, the case of an unstable or strongly convective PBL is not well represented. Thus, an extension for non-local mixing due to large-scale eddies is often invoked in which the counter-gradient term represents large eddy flux and $K$ permits eddy length scales comparable to the PBL height in the case of unstable boundary layers. CarbonTracker

(Peters et al., 2004) and GEOS-Chem (Lin and McElroy, 2010) both use such a nonlocal scheme based on Holtslag and Boville (1993). ECMWF also uses a diagnostic nonlocal scheme (Köhler et al., 2011). In GEM's 1.5 order closure scheme,





$K$ can also represent nonlocal effects due to large eddies in an unstable PBL through a careful definition of eddy length scale (Bélair et al., 1999). One notable difference between GEM's prognostic nonlocal scheme and diagnostic nonlocal schemes is that the latter require PBL height as an input for determining $K$, whereas in GEM's scheme, the PBL height is diagnosed from the turbulent kinetic energy profile (so $K$ is not a direct function of PBL height).

A minimum value for eddy diffusivity of 0.1 $m^2$ $s^{-1}$ is imposed in the PBL in GEM-MACH for air quality applications. This value was empirically chosen to balance the impacts on the various tropospheric reactive species (Paul Makar, 2012, personal communication). In this work, the minimum value was raised to 10 $m^2$ $s^{-1}$ because a value of 1 or 0.1 was found to occasionally lead to too little mixing. This was most evident in spuriously low $CO_2$ concentrations in the daytime during summer when biospheric fluxes are large and negative. (Note that the eddy diffusivity is kept fully varying

in space-- it is only the minimum value that is slightly altered. This is in contrast to earlier versions of GEOS-Chem in which instantaneous full mixing of emissions and mixing ratios occurred in the PBL. See Lin and McElroy, 2010 for example.) A negative consequence of raising the minimum value to 10 is a reduced-amplitude of the diurnal cycle of $CO_2$ (as will be seen in section 3). However, many models have difficulty in capturing the amplitude of the diurnal cycle (Law et al., 2008, Patra et al., 2008) so this consequence was considered tolerable at present. Moreover, the comparison of model results to

measurements on sub diurnal time scales is very difficult, leading Law et al. (2008) to recommend that "comparisons with observations should only be made for daily or longer time averages, and possibly for only part of the diurnal cycle". Thus, many inversion systems assimilate only afternoon mean observations and do not attempt to capture sub diurnal time scales (Peters et al., 2010).

**2.6 The coupled meteorology and tracer transport forecast cycle**

To transport greenhouse gases, a simulation cycle is used. Meteorological analyses archived at the CMC (Canadian Meteorological Centre) are inserted periodically to constrain the transport to reality. While operational assimilation cycles use an update frequency of 6 h, model forecasts during such a short period will be contaminated by spurious gravity wave generation, as evident in surface pressure time series (as well as other fields) (see Daley, 1991, ch. 6). However, after 24 hours, spurious gravity waves have generally dispersed so an update frequency of 24 h was chosen, just as in Agusti-

Panareda et al. (2014). A schematic diagram of the transport cycle is depicted in **Figure 1**. The upper half of the figure depicts the operational system which collects meteorological observations over a 6 h window and uses these in a 4-Dimensional Variational Assimilation or 4D-Var (Gauthier et al., 2007) to generate an estimate of the meteorological state for a deterministic prediction. Although the operational deterministic prediction system now uses a hybrid approach to background error covariance estimation (Buehner et al., 2015), the analyses used here (from 2009-2010) were generated

using the previously operational 4D-Var system (Charron et al., 2012). On the cycle's start date at 0 UTC, the meteorological analysis is combined with an initial state for greenhouse gases (here $CO_2$ only) and a 24 h coupled meteorology and tracer forecast is produced. The 24 h tracer forecast is subsequently combined with the meteorological analysis for the next day at 0 UTC (blue boxes) to produce a new coupled initial state for the second day's forecast (red filled



circles). Note that no additional "initialization" or filtering (see Daley 1991, ch. 6, 9, 10) scheme such as a digital filter or incremental analysis updates (Bloom et al., 1996) is used, as the impact on the $CO_2$ field was found negligible. Also note that in our simulation cycles, no assimilation of greenhouse gases is performed. An assimilation system is currently in development as discussed in section 5.

In **Figure** 1, every time a new analysis is inserted, a new surface pressure field informed by atmospheric observations is introduced. This surface pressure analysis will differ from the 24 h forecast of surface pressure because the model is not perfect, so the model forecast cycle will experience an abrupt shift in global mean surface pressure and hence, global air mass. The change in global air mass will then impact the global tracer mass. To maintain global tracer mass conservation across this discontinuity, it is necessary to redefine the tracer mixing ratio for the change in air mass. The

scheme adopted follows that used by GEM for the global surface pressure adjustment. Specifically, a spatially uniform increment in tracer mixing ratio is determined based on the constraint that the global tracer mass be preserved (see Appendix A for details). The adjustment so obtained is small, with a mean value of $-8\times10^{-5}$ ppm and a standard deviation of 0.015 ppm over a one year simulation. Since the surface pressure analyses are stored with only 16-bit precision (because the analysis uncertainty does not warrant more precision than this), this ultimately limits our knowledge of the tracer mixing ratio and the

local tracer mass. With our global adjustment scheme, the adjustments are much smaller than 16-bit precision will allow, and thus the scheme is justifiable.

## 3  Model evaluation

We will use GEM-MACH-GHG to study the influence of uncertain atmospheric transport on the spatial scales recoverable in $CO_2$ fields, but since this model has not yet been used for greenhouse gas transport, it is necessary to first document its

ability to transport $CO_2$.

### 3.1 Evaluation of meteorological fields

ECCC has been delivering operational weather forecasts for over 40 years and the products have been based on the GEM model for 19 years. As with many national weather forecast centres, ECCC participates in regular intercomparisons of operational forecasts following WMO standards. For the global configuration, forecasts on the medium (up to 10 days)

range are compared every month and are available at
http://web-cmoi.cmc.ec.gc.ca/verification/monthly/observations/obs_monthly_e.html. Thus the quality of GEM transport has been documented. However, since changes were made to the model that affect weather prediction (specifically, the change to the continuity equation, the implementation of conservation of the dry air contribution to the global mean surface pressure, and the convective transport scheme parameters) it is necessary to demonstrate the quality of the meteorological

fields for the purpose of greenhouse gas transport. Since offline transport models often use reanalyses such as ERA-Interim or MERRA to transport constituents, we compare our 24 h forecasts against these products. In general, analyses more



closely match observations than do forecasts because models are imperfect, and reanalyses should be superior in quality to operational analyses because the former use a superset of observations and a single, recent model version. Thus, our 24 h forecasts cannot be expected to be superior to any of ERAI, MERRA or JRA-55 (Kobayashi et al., 2015) reanalyses. Nevertheless, because of the usage of a 24 h forecast cycle (**Figure** 1), it is important to verify that transport on this forecast

range is reasonable, and reanalyses serve as high quality reference fields.

The initial meteorological fields used at the start of every cycle (**Figure** 1) come from archived operational products interpolated to our lower-resolution grid and topography. Thus our transport can only drift from reality for 24 h before it is corrected so that, even with our modifications to GEM, significant transport errors are not expected. Nevertheless, since changes of several hPa in 3-day forecasts of surface pressure do arise from the adjustment to the continuity equation and

these could impact synoptic-scale forecasts, it is worth verifying the quality of our modified transport. **Figure** 2 (left column) shows the monthly mean difference of temperature, zonal and meridional wind differences between GEM-MACH-GHG and ERAI fields for July 2009 based on 6 hourly difference fields. ERAI fields were obtained from http://reanalysis.org on pressure levels in GRIB format and interpolated to GEM's grid at 1.5° resolution. GEM fields were also output on the same pressure levels and resolution. The mean differences can be compared to those between other

reanalyses: MERRA and ERAI (middle column) and JRA-55 and ERAI (right column). The latter difference fields were obtained from http://reanalysis.org using the Web-based Reanalysis Intercomparison Tools (WRIT, Smith et al., 2014) with the values saved in NetCDF format for plotting. It is evident that for all fields, GEM-MACH-GHG is as similar to ERAI as other reanalysis products although our products are 24 h forecasts (not analyses nor reanalyses). For temperature (top row), the largest differences appear in the stratosphere, for all models. In the stratosphere where models and observations are

biased, assimilation can be challenging (Polavarapu and Pulido, 2016) so this result is not surprising. The extratropical zonal mean stratosphere is dominated by the slow Brewer-Dobson circulation which is forced by waves propagating upward from the troposphere (Andrews et al., 1987; Vallis, 2006). The spectrum of waves from a given model depends on parameterizations such as convection and gravity wave drag which generate high frequency subgrid scale waves, as well as resolved waves. So different models can be expected to have rather different spectra, and thus different forcing of the

Brewer-Dobson circulation. For wind fields, largest differences occur in the tropics. Zonal wind differences are greatest in the tropical stratosphere (middle row). The zonal mean tropical stratosphere is dominated by the quasi-biennial oscillation (QBO) which is driven by vertically propagating waves. Climate models have difficulty capturing the QBO as it depends on vertical resolution, gravity wave drag parameterizations and the spectrum of waves generated by tropical convection schemes (Baldwin et al., 2001; Campbell and Shepherd, 2005). Analyses can capture the QBO by assimilating the few

radiosonde wind observations available in the tropics. However, the amplitude and phase of the QBO captured may depend on the underlying model and assimilation system characteristics. In general, since few direct measurements of winds are available to constrain analyses, and there are no simple dynamical balances available to infer winds from observations related to mass fields, model biases in the tropics are difficult to correct with data assimilation (Polavarapu and Pulido,



2016). In **Figure** 2, the meridional winds (bottom row) differ most in the tropical troposphere where these issues of lack of measurements, and simple dynamical balances will prevail.

In summary, the meteorological fields used to transport $CO_2$ in our system are similar in quality to those from reanalyses which have been used with offline transport models. While statistics for July 2009 were shown in **Figure** 2, those
for Dec. 2009, July 2010 and Dec. 2010 are found in the Supplemental Material (**Figs. S3-S5**).

### 3.2 Evaluation of $CO_2$ fields

Having established that the meteorological fields that will transport constituents are sufficiently accurate, the $CO_2$ evolution can be assessed. Since the goal is to assess GEM-MACH-GHG as a transport model, realistic surface-to-atmosphere fluxes are required. Without realistic sources and sinks of $CO_2$, discrepancies with observations could be equally attributed to
erroneous fluxes as to erroneous transport. Even with a good source of retrieved atmospheric fluxes, the evaluation of GEM-MACH-GHG transport requires care (as discussed below). For this purpose, posterior fluxes from CarbonTracker 2013B (hereafter referred to as CT2013B, Peters et al., 2007) and CarbonTracker 2010 (hereafter CT2010) were obtained and regridded in a mass conservative way to GEM's grid. CarbonTracker was chosen because it is generally recognized as a good product, is regularly monitored and updated, and is readily available from http://carbontracker.noaa.gov. These fluxes
were also used for a similar purpose in Houweling et al. (2010). With fluxes from CT2010 and an initial condition from CT2010 for 1 January 2009 0 UTC, GEM-MACH-GHG was run for 2009. With the CT2013B fluxes, GEM-MACH-GHG was run for all of 2009-2010. The resulting $CO_2$ simulations are compared against observations assimilated by CT2010 and CT2013B (continuous observations), observations not assimilated by CT2010 and CT2013B (columns from Total Carbon Column Observing Network (TCCON) and aircraft profiles), and gridded 3-dimensional concentration fields from CT2010
and CT2013B.

**Figure** 3 compares $CO_2$ time series from GEM-MACH-GHG with CT2013B fluxes to surface observations from ECCC's greenhouse gas measurement network (Worthy et al., 2005) at Alert, East Trout Lake and Sable Island. Alert is a remote Arctic site far from $CO_2$ sources so it can be used to assess long-range transport. East Trout Lake is close to sources and will reflect diurnal variations in $CO_2$ fluxes convolved with boundary layer mixing. Sable Island is downstream of
sources on the eastern edge of the continent and effectively reveals synoptic-scale variability. The comparison at Alert (top panel) reveals general agreement between the model and measurements throughout the 2-year period, with the exception of low values during the autumn of both years. This departure can be explained by a discrepancy between CarbonTracker's transport and GEM-MACH-GHG's transport. CT2013B fluxes were obtained by minimizing the difference between observations and CarbonTracker's forecasts and thus reflect the amount of flux needed to bring CarbonTracker in line with
measurements. Since CarbonTracker's transport is not perfect, the retrieved fluxes retain a signature of CarbonTracker's transport errors which may or may not match GEM-MACH-GHG's transport errors. To demonstrate this point, GEM-MACH-GHG was run with another set of retrieved fluxes, this time from GEOS-Chem (Deng et al. 2016) using GOSAT data, for July 2009-Dec 2010 and no such autumnal drift is seen (**Fig. S6**). Thus GEM-MACH-GHG is able to capture long



and seasonal time scales of $CO_2$ transport. The time series at East Trout Lake (middle panel) shows that the smaller amplitude of the diurnal cycle in winter relative to summer is captured by the model. However, CarbonTracker's diurnal cycle amplitude in summer at this site appears too large. The amplitude of the diurnal cycle is further discussed in the context of **Figure 4**. Finally, the bottom panel compares observed and modelled time series at Sable Island where synoptic-

scale variations are more evident. It is clear that the model captures synoptic scales well, as expected from a weather forecast model. In particular, variations on the one month timescale are seen in August and September of 2009 and the model follows these variations well. The ability of GEM-MACH-GHG to capture synoptic scales is important, since the ground-based measurement network can resolve the global carbon budget and very large (continental) spatial scales (e.g. Peylin et al., 2013). Then in a data assimilation or flux inversion system, the model can supplement the large spatial scales

observable with this network with realistic synoptic scales (e.g. Agusti-Panareda et al., 2014).

  The mean diurnal cycle at individual stations can reveal more clearly the realism of a model's boundary layer variation. Here we choose two sites close to sources and sinks because the amplitude of the diurnal cycle at such sites should vary through the year. **Figure 4** shows that the model's mean diurnal cycle at East Trout Lake with GEM-MACH-GHG with CT2013B (blue) or CT2010 (red) fluxes compare well to measurements (black) whereas CT2013B (green) has a

too-large amplitude in April and July 2009 (consistent with **Figure** 3). However, the model's behaviour varies with location and time. At Fraserdale, GEM-MACH-GHG's diurnal cycle amplitude is clearly too low in all months whereas CarbonTracker fares better in July and October 2009. The fact that the diurnal variability from GEM-MACH-GHG is lower than that observed stems from the choice made for the minimum value of eddy diffusivity (section 2.5). Lowering this value can increase the amplitude of the diurnal cycle, but with the increased risk of occasional spuriously low values of $CO_2$ during

summer daytime. As noted earlier, it is difficult to compare models to measurements on sub-diurnal time scales (Law et al., 2008; Patra et al., 2008) and most models have difficulty in capturing boundary layer evolution so flux inversions typically use only afternoon mean measurements. Nevertheless, in the future, this minimum value may be revised as GEM-MACH-GHG's boundary layer is routinely assessed. The fact that the diurnal cycle amplitude is not poor at all locations may make finding a single best parameter value rather difficult. Indeed, Aliabadi et al. (2016) show that the realism of PBL height

estimates in stable conditions compared to observations depends on location as well as the choice of parameterization.

  GEM-MACH-GHG transport is directly compared to CarbonTracker's transport in **Figs.** 5 and 6. **Figure 5** presents the column mean $CO_2$ weighted by air mass on 1 July 2009 0 UTC from GEM-MACH-GHG with CT2013B fluxes (top panel), GEM-MACH-GHG with CT2010 fluxes (middle panel) and CT2010 mole fractions (bottom panel). The largest differences between the simulations occur in the summer, possibly due to the different convection schemes used in the two

models. For example, the GEM-MACH-GHG simulations with different posterior fluxes are closer to each other than to CarbonTracker in the tropical Atlantic. Zonal mean fields also reveal that the greatest differences between the two GEM-MACH-GHG simulations and CT2010 are in the tropics in summer (**Figure 6**, top panel). In winter (**Figure 6**, bottom panel), GEM-MACH-GHG with CT2010 fluxes matches CT2010 in the tropics better than GEM-MACH-GHG with CT2013B fluxes, indicating that the fluxes are more important for explaining the tropical structure than differing model



dynamics at this time. This is consistent with the finding that posterior fluxes are sensitive to prior fluxes particularly in data sparse regions such as the tropics (Peters et al., 2007, 2010). However the differences between both GEM simulations and CT2010 are very small compared to the difference between CT2010 and CT2013B in the tropics (not shown) (which is largely attributed to the convective mass flux fixes in the latter--see

http://www.esrl.noaa.gov/gmd/ccgg/carbontracker/CT2015_doc.php). Differences in the stratosphere, particularly on 31 December 2009 between the two models are also evident in **Figure 6**. Since GEM-MACH-GHG has better vertical resolution compared to CarbonTracker (80 versus 34 levels) and GEM is designed to have a realistic stratosphere (Charron et al., 2012), differences in stratospheric and mesospheric flow are to be expected. Differences at these levels are even greater when compared to CT2013B where vertical resolution was further reduced to 25 levels (not shown). However, the mass of

$CO_2$ in the stratosphere and mesosphere is very small so that column mean or surface values would be insensitive to such differences. In summary, overall, GEM-MACH-GHG with CarbonTracker's retrieved fluxes reproduces CarbonTracker's fields rather well.

To assess seasonal time scales, it is useful to compare to the Total Carbon Column Observing Network (TCCON) measurements (Wunch et al., 2011). The data used for this study are from the GGG2014 release, available on the network's

website http://tccon-wiki.caltech.edu. All sites with measurements in 2009 and 2010 are selected, as listed in **Table 1**. The details of how the model profiles were converted to column-averaged dry mole fraction ($XCO_2$) and smoothed following Wunch et al. (2010) is provided in Appendix B along with the precise definitions of the statistics discussed here (bias, root mean square or RMS, and scatter). The statistics for 2009-2010 are shown in **Table 2**. The bias is below 1 ppm for every station except Eureka which only had 49 hours of measurements in 2010. No selection is applied when considering which

sites are included in the ALL, MEAN or standard deviation (SD) statistics. At Eureka, the -2.66 ppm bias significantly impacts the station-to-station SD of the bias, it is 0.3 ppm without Eureka. Except for Eureka and Park Falls, the station bias is positive and the overall standard deviation is 0.96 ppm with a high correlation coefficient of 0.95. Although we did not perform a data assimilation, the posterior fluxes from CarbonTracker contain information from the observations they used, so we can compare our **Table 2** to Massart et al. (2016, Table 2). The biases of individual stations are mostly lower here, as

is the overall averaged bias. Since the analyses in Massart were based on GOSAT whereas CT2013B used surface data, this may indicate a better agreement of surface observations with TCCON. The full time series for Park Falls (USA) and Wollongong (Australia) are shown in **Figure 7** and the seasonal bias as well the seasonal statistics using data from all sites combined are shown in **Table 3**. Seasons are defined as DJF (December, January, and February), MAM (March, April, and May), JJA (June, July, and August) and SON (September, October, and November). The model is able to reproduce seasonal

variations of $XCO_2$ with biases ranging between -1 and +1 ppm (excluding Eureka) and scatter values consistently below 1 ppm. The large negative bias at Eureka in autumn is consistent with the time series shown in **Figure** 3. Other northern stations (Sodankylä and Bialystok) have similar but smaller biases. As discussed earlier, the discrepancy of GEM-MACH-GHG transport with that of CarbonTracker (as imprinted in the posterior fluxes) explains this behaviour since other posterior fluxes do not have this particular issue (**Fig. S6**).





The vertical structure of model $CO_2$ is compared to NOAA aircraft profiles (Sweeney et al. 2015) over Canada and the U.S. in **Figure 8.** Following Agusti-Panareda et al. (2014), mean model profiles at the nearest model gridpoint to the profile location were averaged over all profiles for a season. Both observed and model values were binned into 1 km layers. The observations are from ObsPack2013 (Masarie et al., 2014) and include only profiles from Canada or the contiguous U.S.

The annually averaged model profiles are shown in panel a. GEM-MACH-GHG has exceptionally good agreement with these independent measurements while CT2013B has a very slight positive bias. Both models are quite good compared to the ensemble of models shown in Stephens et al. (2007, Fig. 2b). However, the other panels reveal that GEM-MACH-GHG's excellent annual result is because of compensating errors in different seasons. The boreal winter season (Dec-Jan-Feb) is not shown because Dec 2008 was not simulated. However, the behaviour of the model in boreal winter 2009 (based on Jan-Feb)

and 2010 is qualitatively similar to its behaviour in spring (panel b). Panels b-d reveal that GEM-MACH-GHG agrees quite well with observations from 3-6 km in all seasons. However, from 1-3 km, vertical gradients are too sharp but of opposing directions in boreal spring (panel b) and summer (panel c). In autumn (panel d) the gradient is slightly too small while CT2013B is almost perfect. Thus, the vertical mixing just above the boundary layer is too weak, as with most models (see Yang et al., 2007). Because the GEM-MACH-GHG profiles obtained with the 2 different posterior fluxes have very similar

profiles in all seasons, the departure of vertical gradients from those observed is attributed to the model's formulation. These biases in vertical gradients will be relevant for regional flux inversions (see Stephens et al. 2007) that may use GEM-MACH-GHG results as boundary conditions. Overall, however, compared to CarbonTracker, vertical gradients in GEM-MACH-GHG agree better with independent measurements in the mid troposphere but less well in the lower troposphere.

**4 The predictability of $CO_2$ in the context of uncertain atmospheric analyses**

Having established that GEM-MACH-GHG can simulate $CO_2$ reasonably well, we turn our attention to the question of how atmospheric transport modulates the $CO_2$ distribution. The evolution of $CO_2$ can be described by the species transport equation and thus may be considered to be perfectly predictable. However, this is only true if the advecting fields are perfectly known, and this is never the case. With a coupled meteorology and forecast model, the impact of the uncertainty of

meteorological fields on $CO_2$ transport can be explored. In a data assimilation or flux inversion system, if the fluxes are perfectly known, the ability to estimate the $CO_2$ fields using observations will ultimately be limited by predictability, just as the quality of weather forecasting products are. Thus it is useful to identify these limits on the spatial scales that can be retrieved in analyses of $CO_2$ because these predictability limits can then be compared to the spatial scales of $CO_2$ retrievable in the presence of meteorological analysis errors. In this section, we use GEM-MACH-GHG to determine the predictability

of $CO_2$ on weather and climate time scales with the latter referring to subseasonal to seasonal scales. In section 4.1, the classic weather predictability problem is considered whereas longer time scales and the transport errors due to uncertain meteorology, model errors and flux errors are considered in section 4.2.





### 4.1 Weather time scales

Although flux inversion systems focus on retrieving relatively long time scale signals (than 2 weeks), it is useful to first consider $CO_2$ predictability on weather time scales before considering errors on longer time scales (next subsection). The predictability problem on weather timescales is related to forecast sensitivity to initial conditions, but the atmospheric

variability of $CO_2$ on diurnal (Law et al., 2008), and seasonal and interannual (Gurney et al., 2002, 2004; Baker et al. 2006a, LeQuéré et al., 2015) time scales is largely governed by the terrestrial biospheric fluxes and hence is determined by sensitivity to boundary rather than initial conditions. Nevertheless, the $CO_2$ predictability error on weather time scales has not to our knowledge been identified, and it can be used to identify an upper limit on forecast errors. This may be relevant for operational data assimilation or forecasting systems such as those at ECMWF (Agusti-Panareda et al. 2014) and NASA

Goddard (Ott et al., 2015) which use update cycles of 12 or 24 h and also examine the quality of short term forecasts. It will also serve as an upper limit for transport error in the presence of uncertain meteorological analyses in section 4.2.

Predictability of weather normally refers to the sensitivity of forecast errors to initial conditions such that any infinitesimal perturbation will lead to diverging forecasts in a finite length of time. This is the so-called butterfly effect and it occurs because of the underlying nonlinear chaotic dynamics of the governing equations (Palmer, 2006). To compute

predictability error of meteorological variables on weather time scales, one can simply start with a reference simulation and perturb the initial conditions. Eventually, the forecasts will diverge, but the error will saturate at climatological levels. Once saturation has been reached, the statistics of this predictability error can be determined. However, with a transported tracer such as $CO_2$, the model forecast requires regular insertion of wind fields. During our forecast cycle, the meteorology is constrained by analyses every 24 h and departure from reality will represent at most a 24 h forecast error. Thus for

transported constituents, the definition of the predictability experiment is slightly different. The reference simulation will be taken as the GEM-MACH-GHG two year run with CT2013B fluxes. Then a comparable "climate cycle" is run in which the model, initial conditions, $CO_2$ fluxes and surface forcing are identical to those used in the reference cycle. However, with the second and all subsequent cycles, the meteorological fields are not replaced by analyses but are instead copied from the 24 h forecast fields. Thus the meteorology fields used to transport the $CO_2$ field in this "climate cycle" are never updated

with observations (analyses) and will thus depart from those used in the reference cycle in the first two weeks. The divergence of the $CO_2$ field in the "climate cycle" from that in the reference cycle, once the error has saturated, defines the $CO_2$ predictability error.

Starting from 1 January 2009 0 UTC, the reference and climate cycles are run for one month with fields saved every 6 h. The differences between the corresponding fields from the two cycles are computed for temperature, zonal and

meridional wind components, $CO_2$ and surface pressure. Vorticity and divergence fields are computed from the wind difference fields. Then the square root of the global mean of the zonal variance of each difference field is computed after first subtracting the zonal mean. The resulting global mean values are called predictability errors and they have the same units as the corresponding forecast field. In order to get a comparable scale for all variables, the errors were normalized by



the square root of the zonal mean variance for a reference state. The choice of this state is arbitrary but has implications on the maximum values attained (as will be discussed below). The reference field was taken as the initial state used to launch both the control and climate cycles and corresponds to 1 January 2009 0 UTC. **Figure 9** shows the time and height variation of the predictability error normalized by the variability of the reference state for all four variables. When the predictability

error approaches the variability of the reference state, values approach 1. Thus predictability is expected only when the relative error is much less than 1. In **Figure 9**, we see that temperature loses predictability within 10 days, as expected (i.e. the normalized error reaches 0.8). However, $CO_2$ loses predictability in the troposphere within 2-3 days except very near the surface where it approaches 5 days. The predictability of $CO_2$ more closely resembles that of the wind (vorticity and divergence) fields which also lose predictability in fewer than 5 days in the troposphere. This makes sense because the wind

fields are used to transport the $CO_2$ field. The difference in evolution of the $CO_2$ field in the reference and climate cycles (figures not shown) reveals largest values to be associated with gradients created by large fluxes (whether natural or anthropogenic). This ability of the uncertainty in wind analyses to act on $CO_2$ gradients and to spread the uncertainty downstream was previously illustrated by Liu et al. (2011) using an ensemble of wind fields. $CO_2$ is more predictable in the stratosphere, with the loss of predictability occurring after 5 days in the lower stratosphere. The extended predictability of

$CO_2$ in the lower stratosphere is similar to that seen in the vorticity field (bottom left panel). The reason that the vorticity field is more predictable than the divergence field (with a loss of predictability occurring after 3-4 days in the troposphere) is because the vorticity field is associated with slower rotational modes whereas the divergence field is often associated with higher frequency waves. The atmospheric kinetic energy spectrum is dominated by rotational motions in the troposphere (Koshyk et al., 1999; Skamarock et al., 2014). In the stratosphere, the zonal mean flow in winter is driven by very large-

scale vertically propagating planetary waves (Andrews et al, 1987; Vallis, 2006) so large-scale rotational modes dominate the energy spectrum and extended predictability in vorticity and $CO_2$ results.

       Stripes of 24 h in frequency are seen in the vorticity and divergence plots. This occurs because the forecasts in the reference cycle are abruptly returned to reality every 24 h with the insertion of a new analysis. When the predictability error of the 24 h forecast error is large compared to that of the analysis valid at the same time, we see stripes at the 24 h period.

Thus we conclude that the normalized 24 h forecast error of the wind field is much larger than that of the temperature field. This makes sense because the global observing system is dominated by information about the mass field with relatively sparse direct observations of the wind field (Baker et al., 2014). In addition, the mass field (which is reflected in the temperature field) is a much smoother field and is thus more easily observable with a given network relative to fields which are dominated by smaller spatial scales (such as vorticity or divergence).

30       While predictability is considered lost when its normalized error approaches 1, sometimes the relative error is much greater than 1. This occurs because of the arbitrary choice taken for the normalization. In our case, the reference state corresponds to the initial state which corresponds to a relatively quiescent synoptic situation since January 2009 marks the strongest and most prolonged stratospheric major warming on record (Manney et al., 2009). The criteria for a stratospheric sudden warming (SSW) were met on 24 January 2009 when zonal mean easterlies replaced the climatologically normal





westerlies at 10 hPa. However, easterlies were noted in the mesosphere prior to this date (Manney et al., 2009). This pattern is consistent with the appearance of anomalously large predictability error in vorticity and divergence in the mesosphere prior to 24 January and the appearance of anomalously large errors in the lower stratosphere on this date. Although global mean values are shown, it is zonal variance that is computed, and the departure from a zonal mean will be large during a

wave 2 vortex splitting event such as occurred in 2009. Thus zonal variances are anomalously large throughout much of the northern hemisphere because the climate cycle does not capture this event whereas the reference cycle does. The extent of the disturbance in the northern hemisphere is large enough to influence the global mean values. Anomalously large $CO_2$ predictability error appears in the mid stratosphere around 20 January 2009. The disturbance of $CO_2$ due to the SSW is to be expected since the disturbance of CO in the mesosphere, $N_2O$ in the mid stratosphere and $H_2O$ in the lower stratosphere as

the vortex deformed and split was evident in MLS (Microwave Limb Sounder) observations (Manney et al., 2009).

         **Figure 10** compares the layer mean normalized predictability error for different variables for the layers indicated by the dashed lines in **Figure 9** and labelled in white text at the left edge of each panel. Near the surface, the normalized $CO_2$ predictability error closely follows that of the specific humidity field (for about 7-8 days). Both moisture and $CO_2$ fields are advected by the wind fields and are similarly affected by the predictability of the wind fields. In this bottom layer, $CO_2$ and

moisture are more predictable than the wind field presumably because of their dependence on surface fluxes. However, they are both less predictable than the temperature field (Fig. 10a). In the troposphere and stratosphere, the loss of $CO_2$ predictability is similar to the loss of predictability in vorticity (panels b,d,e,f). Throughout the atmosphere, temperature loses predictability at a slower rate than does $CO_2$. While both moisture and $CO_2$ are transported by wind fields, moisture is also a dynamic variable thus the loss of predictability for specific humidity is not the same as that for $CO_2$ in the lower to

mid troposphere (panels b,c). Predictability error increases from day 1 to reach saturation levels in 10-15 days for all levels except the upper stratosphere (Fig. 10f) which is being affected by the stratospheric sudden warming. Thus in the next section, climatological levels of predictability error will be discussed for periods longer than 1 month in the troposphere.

         In summary, the global predictability of $CO_2$ on weather time scales in the free troposphere is very short (less than 2 days with our model) and is associated with the predictability of wind fields. This predictability limit refers only to

sensitivity to initial conditions and it will be counter balanced by the predictability coming from biospheric fluxes on diurnal and synoptic scales. However, because atmospheric flow modulates $CO_2$ evolution on these time scales (Parazoo et al., 2008, Law et al., 2008), it is useful to identify these theoretical limits. To improve predictability, more observations will be needed where the wind fields have finer spatial scales and where convection is occurring. The current global meteorological measurement network is relatively sparse in the tropics where convection is important, but new observations from space-

borne lidars may be able to remedy this problem (Baker et al., 2014).

### 4.2 Seasonal time scales

As noted earlier, $CO_2$ observations contain information on seasonal to interannual time scales. Specifically, the global surface network used in flux inversions is able to constrain the global $CO_2$ budget and capture seasonal and interannual





variability of the global fluxes (e.g. Baker et al., 2006a; Peylin et al., 2013). The source of predictability on subseasonal to seasonal and longer time scales partially derives from climate predictability on those scales but also from long time scale information contained in surface fluxes which are, in turn, influenced by climate variability (Patra et al., 2005). In this section, we explore the predictability of $CO_2$ on longer time scales and compare these to $CO_2$ simulation errors due to the use

of uncertain meteorological analyses.

The predictability experiment represents an extreme case in which no information from observations is present in the wind fields after the initial time. In reality, in our $CO_2$ transport cycle (**Figure** 1), the wind fields are constrained to observations by the insertion of a meteorological analysis every 24 h. However, the analyses are not perfect and have a certain level of uncertainty. To simulate this uncertainty, we could perturb the analyses every 24 h with an analysis error. In

a variational data assimilation system such as that used at ECCC, it is possible to estimate the analysis error covariance matrix but it is expensive to do so, and such estimates are not routinely made. On the other hand, a simple perturbation such as random spatially uncorrelated errors will not be useful as they will primarily generate unbalanced motions. What is more relevant is a perturbation of the size and shape of the 6 h analysis error, given the use of a 6 h forecast cycle in operations. (Reanalyses are also available at 6 h intervals and these are sometimes used to constrain flux inversions.) Thus, in order to

simulate a coherent 6 h analysis error, we simply insert the analysis state valid 6 h prior to the actual analysis time (i.e. the one from 18 UTC of the day before, instead of the correct one from 0 UTC), relabelling the date and time to the correct ones. Then, the deviation of the $CO_2$ field from this perturbed analysis cycle from the reference cycle defines the error due to the use of uncertain meteorological analyses. This error should be much smaller than the predictability error.

**Figure 11** shows the monthly mean spatial spectra of various difference fields averaged over several model levels

for July (top row) and December (bottom row) of 2009. The spectra refer to the spherical harmonics (Boer 1983) of a scalar field multiplied by its complex conjugate and summed over zonal wavenumbers. The x-axis then defines a total wavenumber. While spectra were computed for each day at 0 UTC, these were averaged over the month to filter some noise and identify a robust signal. In addition, they were averaged over 12 model levels to get representative spectra for a few atmospheric layers, namely, the bottom 4 layers shown in **Figure 9**. The blue curves in **Figure 11** depict mean spectra of the

$CO_2$ state from the reference cycle similarly averaged in time and in the vertical dimension. The black curves represent the predictability error. This error is very small at the end of the first cycle (on 2 January 2009) but rapidly increases during the first 2 weeks to saturate at climatological levels (section 4.1). Since we are interested in this saturated level of error, we do not consider the first month of errors. The two months chosen in **Figure 11** represent the variation seen in various months of the year. The predictability error is seen to be lower than the reference state itself for very large spatial scales but quickly

equals (around wavenumber 10) then surpasses the power in the reference state. The reason that the power in the predictability error can be larger than that in the state itself is because it involves the difference of two fields. In the limit where two fields become uncorrelated, the variance of the difference equals the sum of the variances. If the two fields have the same climatological variance, the variance of the difference is twice the climatological variance. Thus it is not surprising that the power in the predictability error should surpass that in the reference state for small spatial scales. What is more




intriguing is that some information is still retained in the largest scales (wavenumbers less than 5) even after 6 or 12 months of simulation. The source of this predictability at very large scales is partially from the surface forcing of meteorological fields. Subseasonal to seasonal predictability is manifested in modes of variability such as the Madden-Julian oscillation (MJO), the Pacific North American (PNA) pattern, midlatitude blocking events and the North Atlantic oscillation (NAO)

(Waliser, 2006) and their predictability derives from atmospheric boundary conditions, namely, sea surface temperature, soil moisture, snow cover, vegetation and sea ice (Shukla and Kinter, 2006). These ocean and land surface conditions influence fluxes of moisture, and sensible and latent heat into the atmosphere which may change low level atmospheric convergence and lead to atmospheric heating anomalies which influence the large-scale flow. To see if long time scales in the meteorological analyses explain the large-scale predictability seen in **Figure 11**, an experiment was run in which the

predictability experiment was repeated, but this time incorrect surface fields (from 3 months later) were used. With no information from atmospheric observations as well as a seasonally shifted error in the surface forcing, the predictability error is worsened at these largest scales, particularly in the summer (June, July and August) near the surface (**Fig. S7**), and in the lower troposphere (not shown). The differences in $CO_2$ evolution in the two predictability experiments during boreal summer are largest in the northern extratropics (not shown). This confirms that the ocean and land surface are playing a role

in predictability of the system at the largest scales in the lower troposphere in boreal summer. In the mid and upper troposphere, smaller impacts are seen but the impact is largest in the spring (not shown). In **Fig. S7**, the remaining predictability seen at wavenumbers below 10 for all months is then attributed to the common $CO_2$ fluxes used by the reference and predictability experiments.

In summary, a direct impact of climate predictability on $CO_2$ predictions through the ocean and land surface is seen

through the worsened predictability in boreal summer months when biospheric $CO_2$ fluxes are largest. Since $CO_2$ fluxes were specified, and were the same in the reference and predictability experiments, this climate signal is retained and explains most of the $CO_2$ predictability at large scales in **Figure 11** and **S7**. Finally, it is worth noting that a shift of one month in surface fields resulted in no real deterioration of the predictability error since there is still significant correlation among surface fields due to a one month lag.

The red curves in **Figure 11** depict the 6 h analysis error spectra. As expected, these spectra are reduced compared to predictability errors because meteorological analyses are used in this cycle, although they are 6 h out-of-date. In particular, significant error reduction at large scales is seen. However, the red curves also intersect the reference state spectra at increasingly smaller wavenumbers as height increases. Thus, if analysis errors are considered, there is a gain of information over climatological error levels defined by the predictability error but only for the larger spatial scales. Beyond

the point where the red and blue curves intersect, there is no useful information in the $CO_2$ field at these scales due to meteorological analysis errors. In fact, the $CO_2$ predictions with 6 h analysis errors (red curves) asymptote to the predictability error spectra for large wavenumbers. Near the surface, there is the greatest gain of information, but in the upper troposphere, the spectra of analysis error are less than that of the reference state only for wavenumbers lower than 20. Thus the fact that the meteorological analysis has information on only certain spatial scales places limits on the spatial scales



that can be retrieved in a transported field such as $CO_2$. This result is consistent with the observation that transport biases act at large scales (Chevallier et al. 2010).

The final set of curves (cyan lines) in **Figure 11** depict the spectra of differences in an experiment which is identical to the reference but which uses a different set of posterior fluxes (CT2010 instead of CT2013B). When information differs

in the two sets of retrieved fluxes the spectra of differences are large. It is evident that the differences are greatest at the largest spatial scales and nearer the surface. Since the observing system is broadly similar in the two systems (CT2010 and CT2013B) and both are based on the surface network which defines large spatial scales (e.g. Bruhwiler et al., 2011; Peylin et al., 2013), differences in the two sets of posterior fluxes appear on the largest scales. Where the cyan curves crosses the red curves mark the points where this difference is surpassed by $CO_2$ differences due to analysis error. For example, in the upper

troposphere, the differences produced by using the two different posterior fluxes are always smaller than the uncertainty in $CO_2$ resulting from a 6 h analysis error (layer 4). But near the surface (layer 1), particularly in July, there are large-scale differences resulting from the use of the two different posteriors that exceed the $CO_2$ errors due to the use of imperfect atmospheric analyses. Thus changes to the CarbonTracker system that produced the 2010 and 2013B posterior fluxes are understandably most evident near the surface where the assimilated observations are located and differences are largest in

boreal summer. The difference in $CO_2$ produced by the two products diminishes with height because both CT2010 and CT2013B assimilated surface observations only.

Another view of the spectra can be obtained by summing over total wavenumber and plotting with respect to zonal wavenumber (**Figure 12**) because zonal wavenumber spectra are more indicative of the tropical signal. In this figure, it is evident that there is almost no information retained in the predictability error (i.e. the black curve lies above the blue one).

In other words, predictability error spectra surpass that of the reference state at all levels except near the surface for the first few zonal wavenumbers. When a 3 month shift in surface fields is included in the predictability experiment, all predictability is lost in July and August since the power in the predictability error (red curves) exceeds that of the reference state (black curves) for all wavenumbers (**Fig. S8**). Predictability is mostly lost in September as well. Thus using the correct land and ocean surface fields may be relevant for capturing predictability in July, August and September in the tropics. The

dominant mode of tropical subseasonal variability is the MJO which is characterized by very large scales (zonal wavenumbers 1 and 2 in wind and rainfall fields) in the tropics (Waliser, 2006) and GEM can capture predictability associated with this mode (Lin et al. 2008) supporting the notion that GEM has predictive skill in the tropics on seasonal scales. However, since predictability was already close to lost even with the correct land and ocean surface (power in predictability error is same order of magnitude as that for reference state), this conclusion may not be warranted. Indeed,

outside of these three months, shifting the land and ocean surface fields by 3 months has little impact (red and black curves are similar in **Fig. S8**). This suggests that most of the information retained at the largest scales (seen in **Figure 11**) in the predictability experiment is coming from the extratropics or signals with latitudinal structure for these months. Indeed, the $CO_2$ evolution (not shown) in the reference and predictability experiments differs the most in northern hemisphere extratropics. Thus the large-scale predictability seen in the northern extratropics is due to large $CO_2$ variability during



October to May associated with biospheric fluxes but from June to September, the ocean and land forcing of GEM's climate is also important. When 6 h analysis errors are simulated (red curves), there is a gain of information over predictability error but fewer than 20 zonal waves are resolved in the mid troposphere (**Figure 12**, layers 2 and 3). In the lowest layer, about 40 waves are resolved. Compared to **Figure 11**, there are significantly fewer waves being resolved. Thus structure in the

meridional direction is better resolved in analyses than is structure in the zonal direction. Finally, the differences in the two posterior fluxes (CT2010 and CT2013B) are smaller than $CO_2$ differences due to meteorological analysis errors everywhere, in December. In July, only near the surface and only for the smallest few wavenumbers are differences larger than analysis errors (i.e. cyan curve is above the red curve). Thus, we conclude that, except for July near the surface, the two sets of posterior fluxes have retrieved almost identical information in the tropics.

It is also useful to examine other sorts of model errors in terms of their impact on $CO_2$ predictions. **Figure 13** shows the spectra of errors due to the addition (or not) of convective tracer transport (cyan curves). The impact of adding convective tracer transport is primarily at large scales and exceeds $CO_2$ errors due to imperfect wind analyses (red curves) only in the mid troposphere for wavenumbers less than 5. Spectra for April 2009 were shown in **Figure 13** because the impact of adding convective tracer transport was largest in spring months. The impact is always less than that due to

imperfect wind analyses if zonal wavenumber spectra are considered (bottom row).

## 5 Summary and Discussion

A new capability for simulating $CO_2$ using ECCC's operational weather and environmental prediction models has been developed. The adaptations required for greenhouse gas simulation include the implementation of a global mass fixer for the semi-Lagrangian tracer transport scheme, the implementation of a mixing ratio defined with respect to dry air for tracer

variables, the addition of convective tracer transport and slight modification of a parameter in the boundary layer scheme. A sequence of 24 h meteorological forecasts is used to transport $CO_2$ fields in a forecast cycle involving a coupled meteorological and tracer transport model. Using prescribed posterior fluxes from NOAA's CarbonTracker, the transport of the model has been assessed. The 24 h meteorological forecasts are as similar to the ERA Interim reanalyses as are other reanalyses (MERRA and JRA-55). The $CO_2$ fields compare well to observations assimilated in the posterior fluxes (surface

hourly measurements) as well as to independent observations (TCCON and NOAA aircraft profiles) and to CarbonTracker mole fractions. Synoptic and seasonal timescales are well captured but, as with most transport models, the diurnal cycle amplitude is too low in summer. The vertical gradient in the mid troposphere is excellent but the gradient from 1-3 km does not agree as well with observations and the error in the gradient changes with season.

    Using this coupled meteorological and tracer forecast model (called GEM-MACH-GHG), the influence of

atmospheric transport and the uncertainty of meteorological analyses on $CO_2$ predictions was explored. The predictability of $CO_2$ has been described for the first time. Predictability on weather time scales for $CO_2$ is 2-3 days in the troposphere, reaching 5 days near the surface and in the lower stratosphere for GEM-MACH-GHG. Reduced predictability in the





stratosphere was seen during the prolonged and strong stratospheric sudden warming of January 2009. Predictability of $CO_2$ is shorter than that for temperature and is comparable to the predictability of the wind fields in the free atmosphere. Near the surface, the predictability of $CO_2$ is similar to that of the specific humidity field. Predictability on seasonal timescales was seen at the largest scales and is primarily attributed to long timescales in $CO_2$ fluxes which are in turn partly due to climate

predictability. On top of this, three-month errors in land and ocean surface fields are seen to directly impact $CO_2$ predictability in boreal summer in the northern extratropics. By simulating a 6 h forecast error, meteorological analyses are shown to resolve large scales in $CO_2$ transport, but there is a spatial scale below which there is no information beyond predictability error. By comparing spectra in terms of total and zonal wavenumbers, it is seen that more information is retained in the $CO_2$ field in north-south direction as opposed to the zonal direction. The impact of convective tracer transport

on $CO_2$ fields are largest in boreal spring and exceeds errors due to the use of imperfect atmospheric analyses only in the mid troposphere and only for wavenumbers below 5.

By definition, predictability experiments use a reference simulation against which perturbed simulations are compared. Thus the errors obtained here are system dependent but if the system is representative, then results are likely to be representative of other systems. Indeed, being an operational weather forecast model, GEM is routinely evaluated and we

have shown that 24 h forecasts with our modified version of GEM are comparable to reanalysis products. On the other hand, seasonal predictability is model dependent and is most likely related to model parameterization of subgrid scale processes, particularly, convection (Shukla and Kinter, 2006). Subseasonal predictability is limited by the fact that most models do not capture the MJO (Waliser, 2006). Even if they do capture it, and the initial state has a strong MJO signal, forecast skill may still not be improved because of the complex interplay between MJO and other modes of variability (Lin et al., 2008). Thus,

it may be useful for individual models to be able to characterize $CO_2$ predictability error, particularly on longer time scales, as well as the spatial scales definable in the presence of imperfect meteorological analyses.

The length of time attributed to predictability on weather time scales (e.g. **Figure** 9) is based on a normalized predictability error and an arbitrary threshold value of 0.8. Because of the arbitrariness of the choice of reference state used for normalization, and the threshold value, the length of time for which a variable remains predictable should not be taken as

fundamental. Rather, the important result is the relative time of predictability for the variables. Specifically, the fact that the loss predictability of $CO_2$ follows that of the wind field rather than the temperature field is expected to be a robust result.

Using our experimental design, we have also compared the spatial scales of the impact of using different posterior fluxes (from CT2010 and CT2013B) to those arising from the impact of imperfect meteorological analyses. The information of observations is present in the posterior fluxes but since both sets of posteriors used very similar observing

systems, little difference was seen in the resulting $CO_2$ simulations. However, if prior fluxes are used to define a reference simulation then various posterior fluxes could be used to define flux analysis increments. Specifically, the spatial scales resolved by various observing systems (e.g. GOSAT or OCO-2 versus the surface network) could be computed and compared to the scales resolvable in the context of imperfect meteorological analyses. Indeed, such experiments are in progress and will be described in a subsequent article.



The impact of uncertain meteorological analyses on the spatial scales resolved in $CO_2$ simulations was presented here in the context of a coupled meteorological and tracer forecast model. With an offline transport model, the impact of uncertain atmospheric analyses could also be considered. For example, when integrating posterior fluxes to get $CO_2$ concentrations, two simulations could be performed: one with the correct analyses and another with analyses shifted by 6 h.

In addition, the fluxes retrieved using the correct atmospheric analyses could be compared to those retrieved using a series of temporally shifted atmospheric analyses. The difference in the retrieved fluxes in these experiments might then shed light on the magnitude of the impact of transport error due to atmospheric analyses on retrievals.

The impact of uncertain meteorological analyses on $CO_2$ simulations was assessed using a 6 h shift in analysis states as a proxy for 6 h analysis errors. This was done because such estimates of meteorological analysis errors are not available

but a perturbation of the size and shape of a 6 h analysis error was desired. While other proxies for 6 h analysis errors could be devised, none would be any more valid. On the other hand, it is possible to directly obtain analysis and forecast errors by implementing an ensemble Kalman filter for $CO_2$ state estimation. Indeed, a greenhouse gas data assimilation system based on an augmented state (meteorology, constituent, and fluxes) ensemble Kalman filter is now under development. This new system is called EC-CAS (ECCC Carbon Assimilation System) and also uses existing tools developed at ECCC, namely, the

operational global ensemble prediction system (Houtekamer et al., 2014). With EC-CAS, the atmospheric modulation of $CO_2$ forecast uncertainty would be directly simulated and the impact on flux estimate uncertainties could then be determined.

## 6 Author contributions

SP designed, performed and analysed the experiments and wrote the manuscript. SP and MN developed and diagnosed the GEM-MACH-GHG model from 2011-15 and MN developed or contributed to all of the model diagnostics. MT

implemented the tracer mass conservation scheme in GEM, while MT and CG devised and implemented the fixes for the continuity equation and the implementation of the dry tracer mixing ratio variable. DC helped to analyse model results during the implementation of the dry tracer mixing ratio variable. JdeG and SG developed the GEM-MACH model on the global domain and provided input on the implementation and debugging of GEM-MACH-GHG. KSem implemented the convective tracer transport scheme in GEM-MACH. SRen contributed to GEM-MACH-GHG model development during

2011-14 and evaluated its PBL. SRoc and KStr diagnosed model improvements through comparisons to TCCON measurements.

## Acknowledgements

We are grateful to D.B.A. Jones for his encouragement and support throughout the course of this work (over 5 years). We thank Stéphane Bélair for helpful discussions and Bakr Badawy for comments on an earlier version of this manuscript.

Funding from the Canadian Space Agency (CSA) during 2011-14 supported the work of S. Ren through an MOU with



ECCC as well as that of M. Neish through a grant led by D.B.A. Jones of the University of Toronto. Comparisons with TCCON by S. Roche were supported by the CAFTON project, funded by the CSA's FAST Program. We would like to thank Doug Worthy of Atmospheric Science and Technology Directorate (ASTD), Environment and Climate Change Canada, for developing, and maintaining ECCC's greenhouse gas measurement network and for providing the CO2 concentration

measurement data. TCCON data were obtained from the TCCON Data Archive, hosted by the Carbon Dioxide Information Analysis Center (CDIAC) at http://tccon.ornl.gov/ . We thank TCCON PIs Paul Wennberg, Caltech (Lamont, Park Falls), David Griffith, University of Wollongong (Darwin and Wollongong), Justus Notholt, University of Bremen (Bremen), Nicholas Deutscher, University of Bremen (Bialystok), Thorsten Warneke, University of Bremen (Orleans), Dave Pollard, NIWA (Lauder), Ralf Sussmann, IMK-IFU (Garmisch), Kimberly Strong, University of Toronto (Eureka), Rigel Kivi, FMI

(Sodankylä), Frank Hase, KIT (Karlsruhe), and Matthias Schneider, KIT (Izaña). We are grateful to Colm Sweeney (NOAA ESRL) for providing the NOAA aircraft profiles and to Ken Masarie (NOAA ESRL) for compiling ObsPack2013. The National Oceanic and Atmospheric Administration (NOAA) North American Carbon Program has funded NOAA/ESRL Global Greenhouse Gas Reference Network Aircraft program. Finally, we are grateful to Andy Jacobson and to NOAA for the availability of CarbonTracker model products which were invaluable to us when developing our model. CarbonTracker

CT2010 and CT2013B results were provided by NOAA ESRL, Boulder, Colorado, USA from the website at http://carbontracker.noaa.gov .

### 7 Appendix A: Adjusting tracer mass due to changes in surface pressure

Every 24 h when a new surface pressure analysis becomes available, there will be a sudden change in local and global mean surface pressure when the 24 h forecast is replaced by the analysis. Here we derive the scheme used to ensure that the tracer

mass is not affected by this change in surface pressure. First we define some global mass quantities. The global dry air mass is

$$M^a_{dry-air} = \frac{1}{g} \sum_{i,j,k} \Delta p^a_{i,j,k} \left(1 - q^a_{i,j,k}\right)\Delta A_{i,j} \quad , \tag{A1}$$

the global mass of tracer in the forecast is

$$M^f_c = \frac{1}{g} \sum_{i,j,k} c^f_{i,j,k} \Delta p^f_{i,j,k} \left(1 - q^f_{i,j,k}\right)\Delta A_{i,j} \tag{A2}$$

and the global mass of tracer after adjustment is given by

$$M^a_c = \frac{1}{g} \sum_{i,j,k} c^a_{i,j,k} \Delta p^a_{i,j,k} \left(1 - q^a_{i,j,k}\right)\Delta A_{i,j} \quad . \tag{A3}$$

Here $i,j,k$ are longitude, latitude and vertical grid indices, $\Delta A_{i,j}$ is the area of grid box $(i,j)$, $\Delta p_{i,j,k}$ is the vertical pressure difference across the grid box at level $k$, $c_{i,j,k}$ is the mass mixing ratio with respect to dry air at the center of grid box $(i,j,k)$, $q$




is specific humidity and $g$ is the gravitational constant. The superscripts $a$ and $f$ refer to the analysis and forecast, respectively. We seek a spatially invariant adjustment ($\varepsilon$) to the tracer mixing ratio:

$$c_{i,j,k}^{a} = c_{i,j,k}^{f} + \varepsilon \ . \tag{A4}$$

The adjustment parameter ($\varepsilon$) is determined from the constraint that the global adjusted tracer mass equal that of the forecast

tracer. (A4) has the nice property of exactly conserving spatial gradients. For a tracer with large background value (like $CO_2$) this adjustment is much smaller than field itself, and is neglegible when comparing to analysis errors. However, this additive adjustment scheme may have undesirable effects for tracers with a large dynamic range. Substituting (A4) into (A3) yields

$$M_{c}^{a} = M_{c}^{*} + \varepsilon M_{dry-air}^{a} \tag{A5}$$

where

$$M_{c}^{*} = \frac{1}{g} \sum_{i,j,k} c_{i,j,k}^{f} \Delta p_{i,j,k}^{a} \left(1 - q_{i,j,k}^{a}\right) \Delta A_{i,j} \ . \tag{A6}$$

Solving for $\varepsilon$ with the constraint that $M_{c}^{a} = M_{c}^{f}$ yields

$$\varepsilon = \frac{M_{c}^{f} - M_{c}^{*}}{M_{dry-air}^{a}} \ . \tag{A7}$$

**8 Appendix B: Model comparisons to TCCON observations**

For a fair comparison to TCCON observations, GEM-MACH-GHG $XCO_2$ simulations are smoothed using the TCCON a priori profiles and averaging kernels to account for the sensitivity of the measurements. The vertical column (VC) of $CO_2$ is defined as:

$$VC_{CO_2} = \sum_{i=1}^{N} \frac{f_{CO_2,i}}{g \, m_{air,i}} \Delta p_i \tag{B1}$$

with $N$ being the number of layers in the profile, $f_{CO2,i}$ the mean mole fraction in the $i^{th}$ layer of pressure thickness $\Delta p_i$ and

mean molecular weight of air $m_{air,i}$. $g$ is the Earth acceleration due to gravity and is kept constant ($g$=9.80616 m.s$^{-2}$). GEM-MACH-GHG produces profiles of $CO_2$ dry mole fraction $f_{CO_2}^{dry} = \frac{f_{CO_2}}{1 - f_{H_2O}}$ as well as specific humidity $q = \frac{[kg_{H_2O}]}{[kg_{air}]}$. Considering $\frac{1 - f_{H_2O}}{m_{air}} = \frac{1 - q}{m_{air}^{dry}}$ we can express (B1) as:

$$VC_{CO_2} = \sum_{i=1}^{N} \frac{(1 - f_{H_2O,i}) f_{CO_2,i}^{dry}}{g \, m_{air,i}} \Delta p_i = \sum_{i=1}^{N} \frac{(1 - q_i) f_{CO_2,i}^{dry}}{g \, m_{air,i}^{dry}} \Delta p_i \tag{B2}$$



and the vertical column weighted by the TCCON column averaging kernels is :

$$VC_{CO_2,a} = \sum_{i=1}^{N} \frac{(1-q_i) f_{CO_2,i}^{dry} a_i}{g\, m_{air,i}^{dry}} \Delta p_i \ . \tag{B3}$$

The column-averaged dry mole fraction $XCO_2$ is the ratio of the vertical column of $CO_2$ and the vertical column of dry air. Finally the smoothed GEM-MACH-GHG $XCO_2$ is obtained as:

$$X_{CO_2}^{model} = X_{CO_2}^{a\,priori} + \frac{VC_{CO_2,a}^{model} - VC_{CO_2,a}^{a\,priori}}{VC_{air}^{dry}} \ . \tag{B4}$$

GEM-MACH-GHG $CO_2$ profile simulations extend from 2009 to 2010 with an output frequency of 15 min. Average hourly $XCO_2$ is considered for the comparisons and all TCCON sites with observations in 2009-2010 are used. There are both 120HR and 125HR measurements at Lauder in 2009-2010, with the 125HR dataset starting 2 February 2010; only the 125HR data is used. Several statistical parameters are derived: N is the number of pairs (hours) for which there are TCCON measurements. The Bias is the average difference between the model and TCCON:

$$Bias = \frac{\sum_{i=1}^{N} \left( X_{CO_2}^{model} - X_{CO_2}^{TCCON} \right)_{(i)}}{N} \ . \tag{B5}$$

The root-mean-square (RMS) of the differences is :

$$RMS = \sqrt{\frac{\sum_{i=1}^{N} \left( X_{CO_2}^{model} - X_{CO_2}^{TCCON} \right)_{(i)}^{2}}{N}} \ . \tag{B6}$$

The Scatter is the standard deviation of the differences :

$$Scatter = \sqrt{\frac{\sum_{i=1}^{N} [\left( X_{CO_2}^{model} - X_{CO_2}^{TCCON} \right)_{(i)} - Bias]^2}{N-1}} \ . \tag{B7}$$

The mean bias is, with $N_s$ the number of stations :

$$Bias_{mean} = \frac{1}{N_S} \sum_{i=1}^{N_S} Bias_{(i)} \ . \tag{B8}$$

The standard deviation of a station 's bias is useful for estimating the variability of the bias from station to station :

$$SD = \sqrt{\frac{\sum_{i=1}^{N_S} (Bias_{(i)} - Bias_{mean})^2}{N_S - 1}} \ . \tag{B9}$$

20   Finally, R is the Pearson's correlation coefficient :



$$R = \frac{\sum_{i=1}^{N}(X_{CO_2\ (i)}^{model} - X_{CO_2\ mean}^{model})(X_{CO_2\ (i)}^{TCCON} - X_{CO_2\ mean}^{TCCON})}{\sqrt{\sum_{i=1}^{N}(X_{CO_2\ (i)}^{model} - X_{CO_2\ mean}^{model})^2}\sqrt{\sum_{i=1}^{N}(X_{CO_2\ (i)}^{TCCON} - X_{CO_2\ mean}^{TCCON})^2}} \ . \tag{B10}$$

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

**Table 1**: TCCON stations used, their location and data reference.

| Site | Latitude | Longitude | Reference |
|---|---|---|---|
| Eureka | 80.05 | -86.42 | Strong et al. (2014) |
| Sodankylä | 67.37 | 26.63 | Kivi et al. (2014) |
| Bialystok | 53.23 | 23.02 | Deutscher et al. (2014) |
| Bremen | 53.10 | 8.85 | Notholt et al. (2014) |
| Karlsruhe | 49.10 | 8.44 | Hase et al. (2014) |
| Orléans | 47.97 | 2.11 | Warneke et al. (2014) |
| Garmisch | 47.48 | 11.06 | Sussmann and Rettinger (2014) |
| Park Falls | 45.94 | -90.27 | Wennberg et al. (2014a) |
| Lamont | 36.60 | -97.49 | Wennberg et al. (2014b) |
| Izaña | 28.30 | -16.48 | Blumenstock et al. (2014) |
| Darwin | -12.43 | 130.89 | Griffith et al. (2014a) |
| Wollongong | -34.41 | 150.88 | Griffith et al. (2014b) |
| Lauder | -45.05 | 169.68 | Sherlock et al. (2014) |





**Table 2: Statistics for the average hourly XCO2 comparison between TCCON measurements and GEM-MACH-GHG simulations: RMS (ppm), Bias (ppm), Scatter (ppm), and correlation coefficient R. The « ALL » line shows the mean of each parameter using data from all sites combined. MEAN is the average of each parameter and SDis the standard deviation of the station 's bias. N is the number of data pairs (or sites for MEAN and SD) used in the computation of the statistics.**

| Site | Latitude | N | RMS (ppm) | Bias (ppm) | Scatter (ppm) | R |
|---|---|---|---|---|---|---|
| Eureka | 80.05 | 49 | 2.86 | -2.66 | 1.08 | 0.77 |
| Sodankylä | 67.37 | 1384 | 1.23 | 0.19 | 1.22 | 0.98 |
| Bialystok | 53.23 | 1279 | 1.16 | 0.55 | 1.03 | 0.95 |
| Bremen | 53.10 | 455 | 1 | 0.48 | 0.88 | 0.95 |
| Karlsruhe | 49.10 | 274 | 1.42 | 0.95 | 1.05 | 0.91 |
| Orléans | 47.97 | 910 | 0.76 | 0.27 | 0.71 | 0.98 |
| Garmisch | 47.48 | 1194 | 1.1 | 0.22 | 1.08 | 0.93 |
| Park Falls | 45.94 | 2427 | 0.97 | -0.2 | 0.95 | 0.97 |
| Lamont | 36.60 | 4490 | 0.87 | 0.01 | 0.87 | 0.94 |
| Izaña | 28.30 | 221 | 1.13 | 0.45 | 1.04 | 0.84 |
| Darwin | -12.43 | 1704 | 0.67 | 0.28 | 0.61 | 0.88 |
| Wollongong | -34.41 | 1451 | 0.89 | 0.38 | 0.8 | 0.8 |
| Lauder | -45.05 | 826 | 0.58 | 0.35 | 0.47 | 0.88 |
| ALL | | 16615 | 0.98 | 0.14 | 0.96 | 0.95 |
| MEAN | | 14 | 1.13 | 0.1 | 0.9 | 0.91 |
| SD | | 14 | | 0.84 | | |





**Table 3: Seasonal bias (ppm) for the hourly averaged XCO2 comparison between TCCON measurements and GEM-MACH-GHG simulations. The mean (using data from all sites) bias (ppm), scatter (ppm) and correlation coefficient (R) are also shown under « ALL ». Seasons with less than 10 pairs are not included in the « ALL » calculations.**

| Site | Latitude | N | Bias (ppm) | N | Bias (ppm) | N | Bias (ppm) | N | Bias (ppm) |
|---|---|---|---|---|---|---|---|---|---|
| | | | DJF | | MAM | | JJA | | SON |
| Eureka | 80.05 | 0 | - | 0 | - | 35 | -2.18 | 14 | -3.86 |
| Sodankylä | 67.37 | 56 | 0.83 | 450 | 1.04 | 610 | 0.06 | 262 | -1.04 |
| Bialystok | 53.23 | 92 | 0.38 | 557 | 0.82 | 537 | 0.49 | 93 | -0.61 |
| Bremen | 53.1 | 34 | 0.72 | 237 | 0.72 | 123 | 0.02 | 61 | 0.36 |
| Karlsruhe | 49.1 | 5 | 1.32 | 59 | 1.09 | 113 | 0.96 | 97 | 0.85 |
| Orléans | 47.97 | 73 | -0.27 | 310 | 0.38 | 236 | 0.66 | 291 | -0.03 |
| Garmisch | 47.48 | 107 | 0.2 | 254 | 0.4 | 439 | 0.29 | 394 | 0.03 |
| Park Falls | 45.94 | 311 | -0.22 | 668 | -0.13 | 676 | -0.42 | 772 | -0.06 |
| Lamont | 36.6 | 850 | -0.03 | 983 | -0.05 | 1602 | -0.16 | 1055 | 0.34 |
| Izaña | 28.3 | 25 | -0.65 | 44 | -0.56 | 103 | 0.76 | 49 | 1.28 |
| Darwin | -12.43 | 310 | -0.04 | 391 | 0.1 | 126 | 0.5 | 877 | 0.44 |
| Wollongong | -34.41 | 445 | 0.11 | 185 | 0.42 | 332 | 0.96 | 489 | 0.23 |
| Lauder | -45.05 | 168 | 0.01 | 142 | 0.39 | 192 | 0.44 | 324 | 0.45 |
| ALL | | | | | | | | | |
| N | | | 2471 | | 4280 | | 5124 | | 4778 |
| Bias (ppm) | | | 0.02 | | 0.33 | | 0.12 | | 0.15 |
| Scatter (ppm) | | | 0.78 | | 0.88 | | 1.11 | | 0.86 |
| R | | | 0.93 | | 0.92 | | 0.94 | | 0.93 |





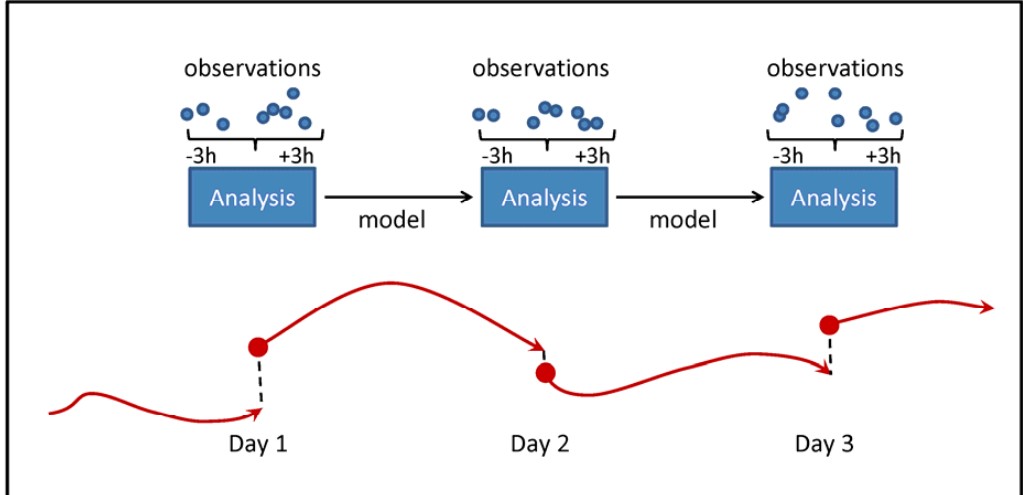

**Figure 1:** Schematic diagram of EC-CAS forward model cycles. Meteorological analyses were precomputed by CMC's operational global deterministic prediction system using all observations collected in a 6 h window centred on the analysis time. These analyses are represented by blue boxes. $CO_2$ tracers are added to these analyses to form an initial condition (solid red circles) for launching a 24 h forecast (red arrows) using the coupled meteorological/tracer model starting at 0 UTC of each day. Note that the meteorological update cycle is 6 h so there should be 4 times as many blue boxes as shown. Some of these were omitted for clarity of presentation.



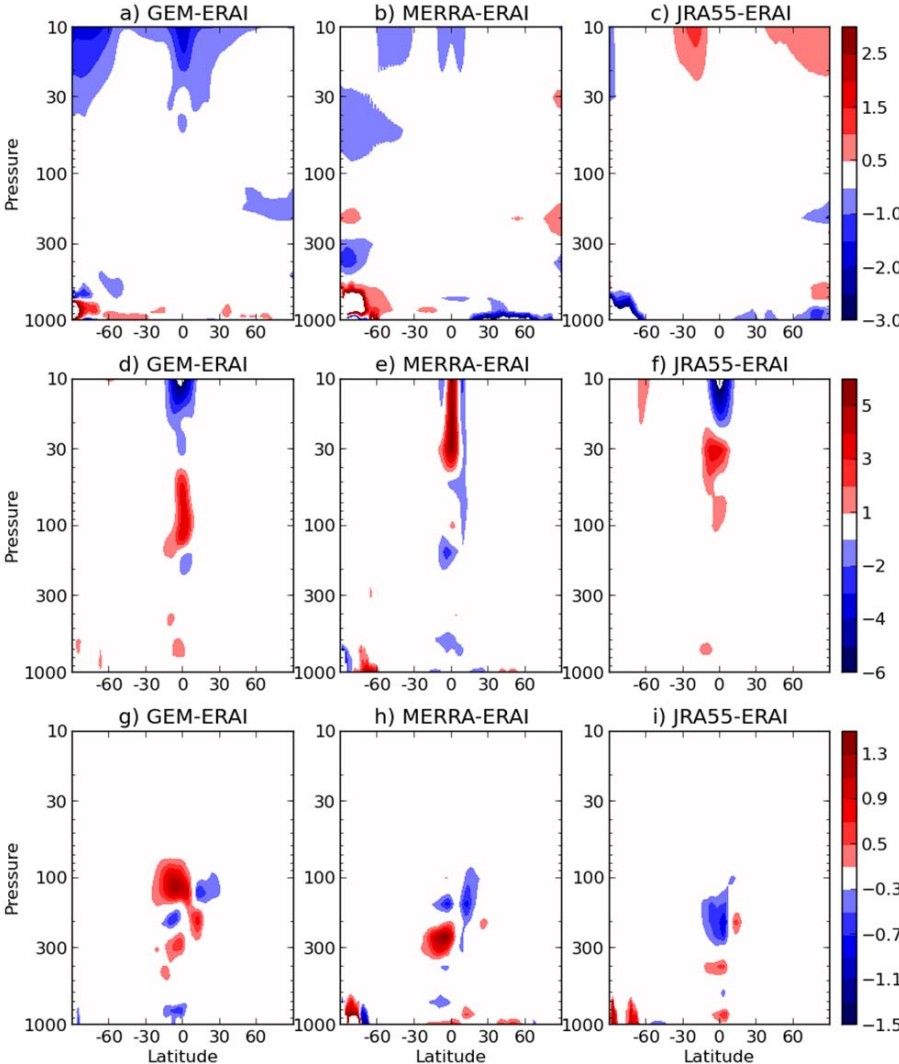

**Figure 2: Comparison of GEM-MACH-GHG meteorological analyses with other reanalyses for July 2009. Monthly and zonal means of differences with respect to ERA-Interim fields of GEM-MACH-GHG (left column), MERRA (middle column) and JRA55 (right column) are shown for temperature in K (top row), zonal wind in m/s (middle row) and meridional wind in m/s (bottom row).**



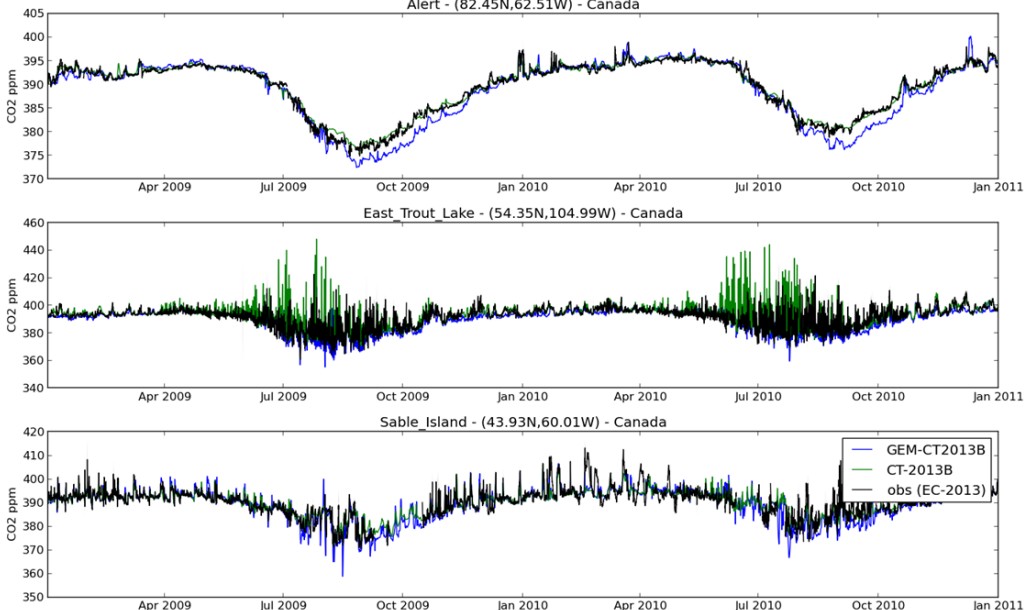

**Figure 3: Comparison of GEM-MACH-GHG with surface observations at Alert (top), East Trout Lake (middle) and Sable Island (bottom). ECCC observations (black), GEM-MACH-GHG with CT2013B fluxes (blue) and CarbonTracker-2013B (green) time series are shown for each location.**




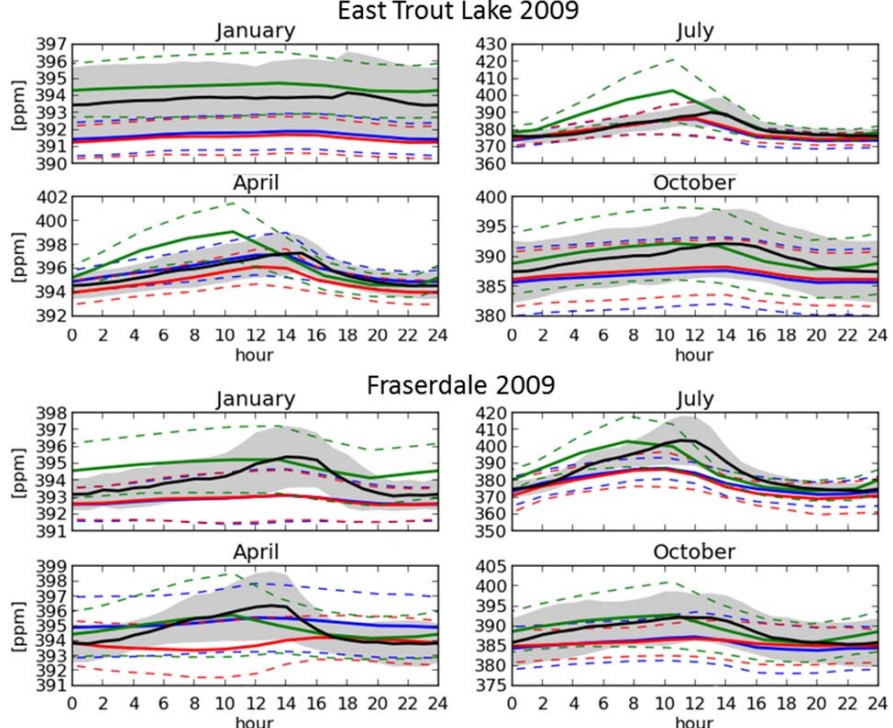

**Figure 4: Mean diurnal cycle at East Trout Lake, Saskatchewan (top 4 panels) and Fraserdale, Ontario (bottom 4 panels) in 2009. Each panel shows the observed mean cycle from continuous measurements (black), CT2013B (green), GEM-MACH-GHG with CT2013B fluxes (blue) and GEM-MACH-GHG with CT2010 fluxes (red). Time is given in UTC. The 4 panels correspond to the months of January, July, April and October, as labelled above each panel. The grey shaded region indicates one standard deviation above and below observed values while the dashed lines indicate the same for the model run with the corresponding colour.**





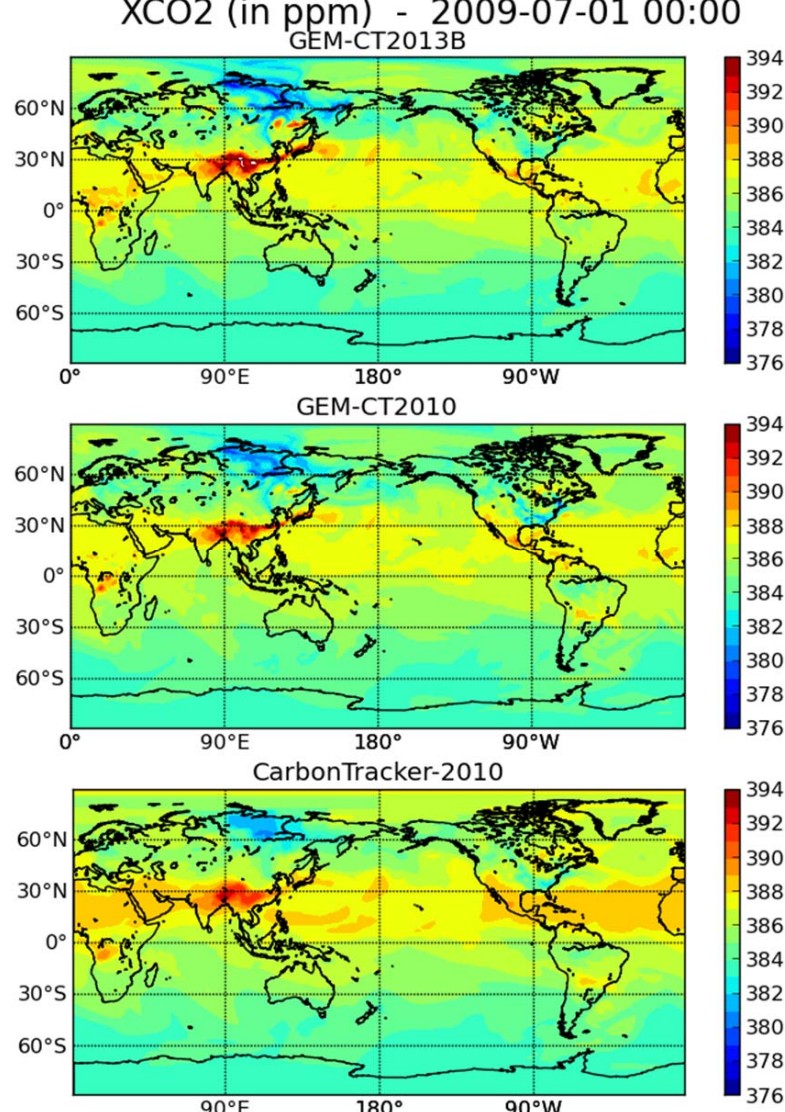

**Figure 5: Column mean CO2 on 1 July 2009 for GEM-MACH-GHG with CT2013B fluxes(top), GEM-MACH-GHG with CT2010 fluxes (middle) and CT2010 (bottom).**





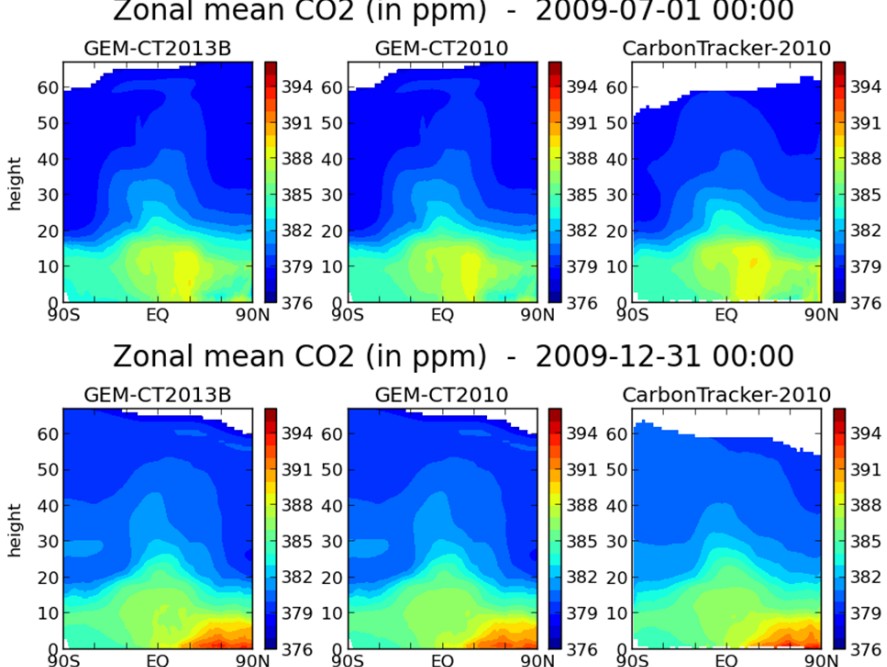

**Figure 6: Zonal mean CO2 on 1 July 2009 (top row) and 31 Dec. 2009 (bottom row) for GEM-MACH-GHG with CT2013B fluxes (left column), GEM-MACH-GHG with CT2010 fluxes (middle column), and CT2010 fields (right column).**





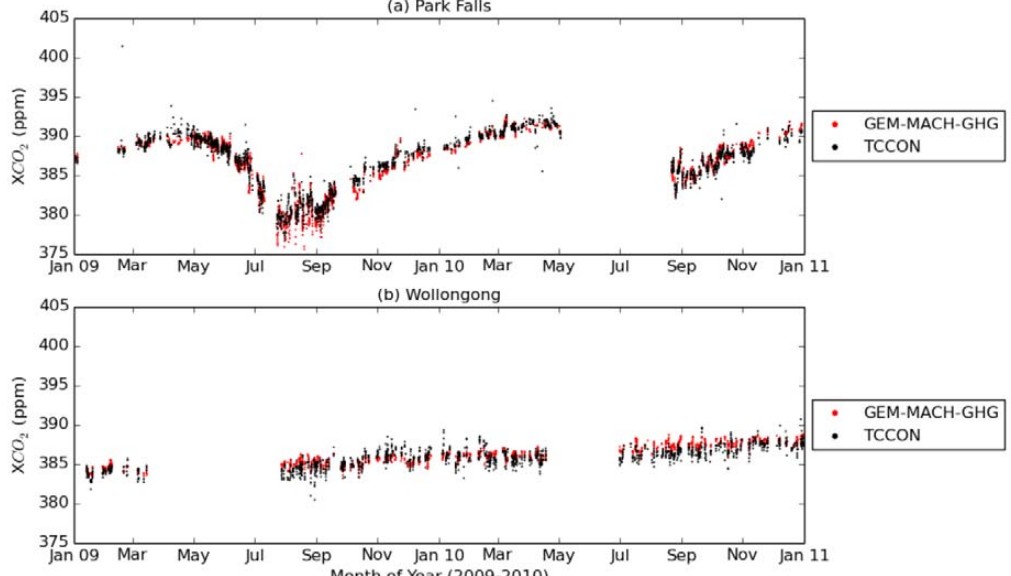

**Figure 7:** **Time series of XCO2 (ppm) at (a) Park Falls, USA and (b) Wollongong, Australia, between January 1, 2009 and January 1, 2011. The dots are hourly averaged XCO2 for TCCON (black) and smoothed GEM-MACH-GHG simulations obtained using CT2013B posterior fluxes (red).**





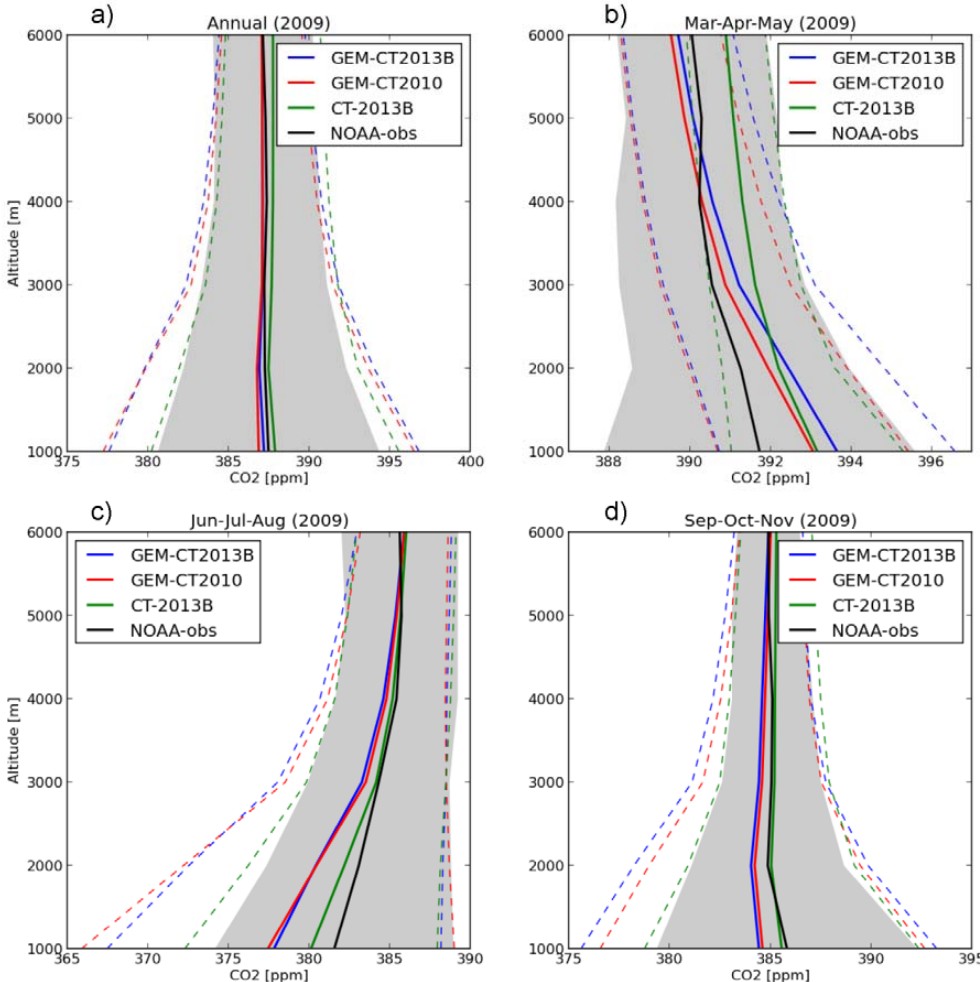

**Figure 8: Comparison of model mean profiles to NOAA aircraft observations.** Observations (black curves) are from obspack_co2_1_PROTOTYPE_v1.0.4_2013-11-25 for locations over continental U.S. and Canada, only. Observed and modelled profiles are binned over (a) 2009, (b) March to May 2009, (c) June to August 2009, (d) Sept. to Nov. 2009. CarbonTracker 2013B
5  molefractions (green), GEM-MACH-GHG with CT2013B posterior fluxes (blue curves) and GEM-MACH-GHG with CT2010 posterior fluxes (red curves) are shown in all panels. The shaded grey regions indicate plus or minus one standard deviation for the observations while the dashed coloured lines indicate the same quantities but for the different model runs. Note that the x-axis range differs in each panel and that ticks are every 5 ppm except for panel b where they are every 2 ppm. Sites used are: Beaver Crossing, Nebraska; Bradgate, Iowa; Briggsdale, Colorado; Cape May, New Jersey; Charleston, South Carolina; Dahlen, North
10 Dakota; East Trout Lake, Saskatchewan; Estevan Point, British Columbia; Fairchild, Wisconsin; Harvard Forest, Massachusetts; Homer, Illinois; Oglesby, Illinois; Park Falls, Wisconsin; Poker Flat, Alaska; Sinton, Texas; Southern Great Plains, Oklahoma; Trinidad Head, California; West Branch, Iowa; Worcester, Massachusetts.



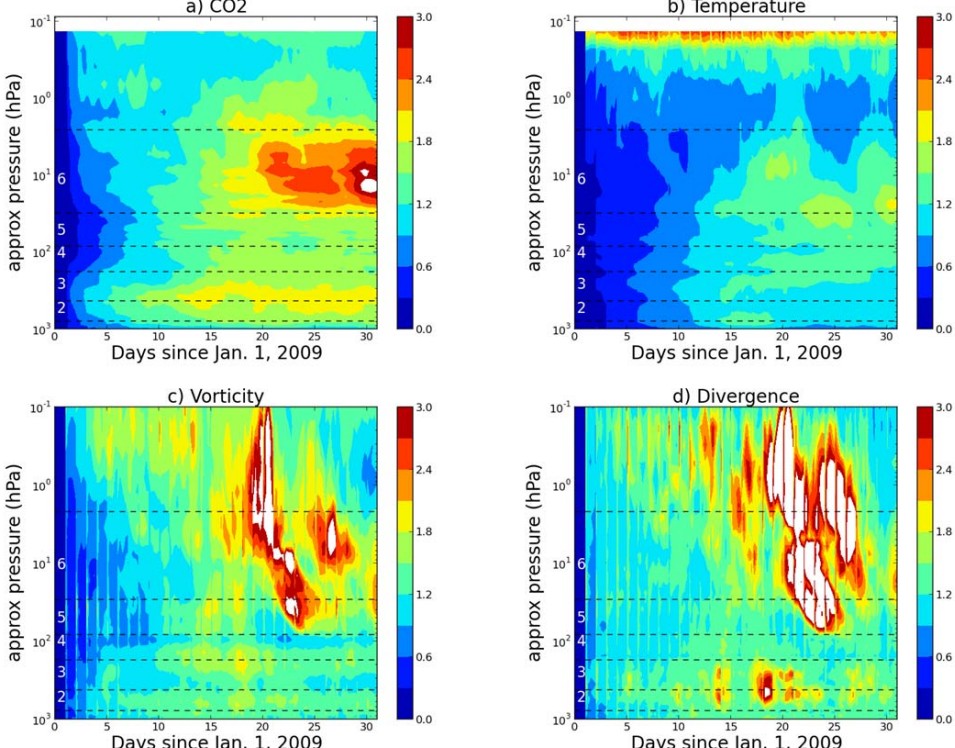

**Figure 9: Predictability on weather time scales during January 2009. Predictability error is defined here as the square root of the global mean variance of the difference in evolution of a control cycle from a climate cycle. This value is normalized by the global mean variance of the corresponding field in the initial condition. Predictable regimes are ones for which this ratio is much less than 1. The normalized predictability error fields for a) CO2, b) temperature, c) vorticity and d) divergence are plotted as a function of model vertical level (converted to approximate pressure with a reference surface pressure of 1000 hPa) and time in days since 1 January 2009. The layers labelled with white text in each panel will be used for computing the layer mean averages shown in Figure 10.**




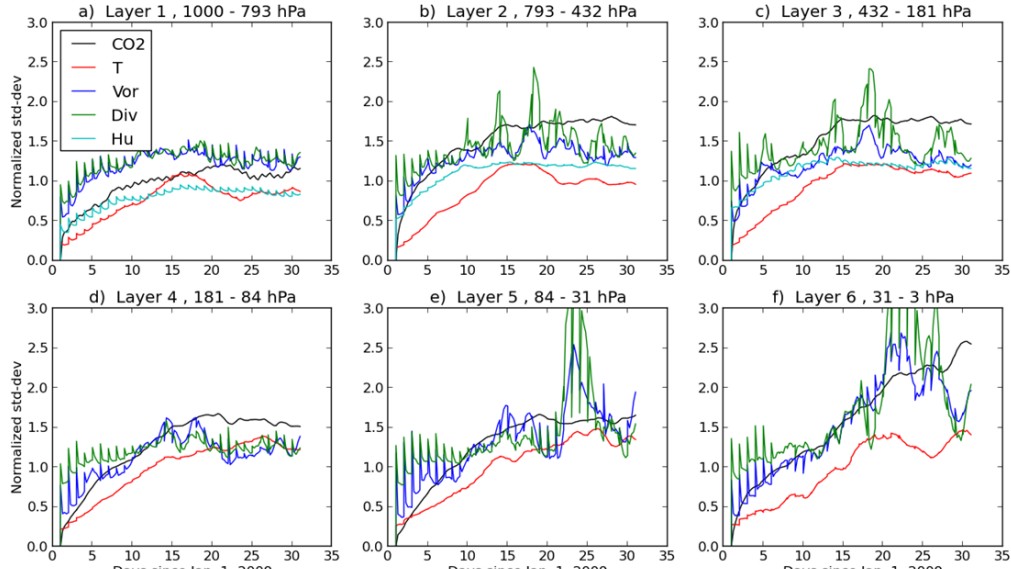

**Figure 10: Time series of layer mean normalized predictability error during January 2009 for CO2 (black curves), temperature (red curves), vorticity (blue curves), divergence (green curves) and specific humidity (cyan curves). For clarity, no curves for specific humidity are plotted in the bottom row of panels. The normalized errors are averaged over 12 model levels, the top and bottom levels used in the average are given in approximate pressure above each panel. The layer numbers are associated with the layers defined by dashed horizontal lines in Figure 9 and correspond to the near surface (layer 1, panel a), the lower troposphere (layer 2, panel b), the mid troposphere (layer 3, panel c), the upper troposphere (layer 4, panel d), the lower stratosphere (layer 5, panel e) and the upper stratosphere (layer 6, panel f).**





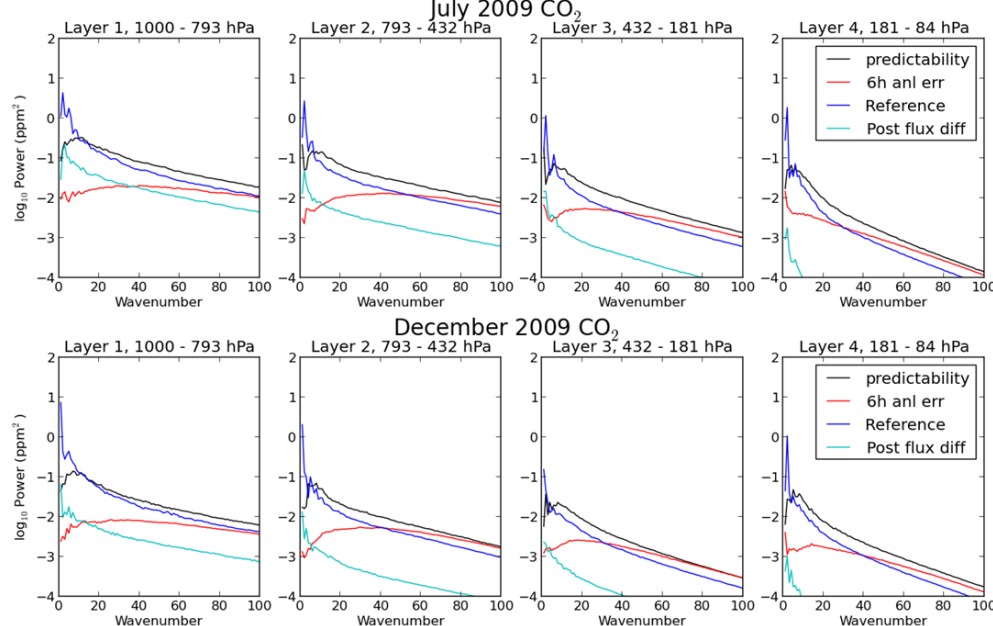

**Figure 11: Spectra of various fields as a function of total wavenumber. Spectra are averaged over one month for July 2009 (top row) and December 2009 (bottom row) and over 12 model levels. The lower and upper model levels averaged are indicated above each frame. Approximate pressure is obtained from model level by multiplying by 1000 (which corresponds to assuming a reference surface pressure of 1000 hPa). The CO2 reference state spectra (blue curves), predictability error (black curves), error due to a 6 h shift in analysis fields (red curves) and differences due to the use of different posterior fluxes (CT2010 or CT2013B) (cyan curves) are shown.**





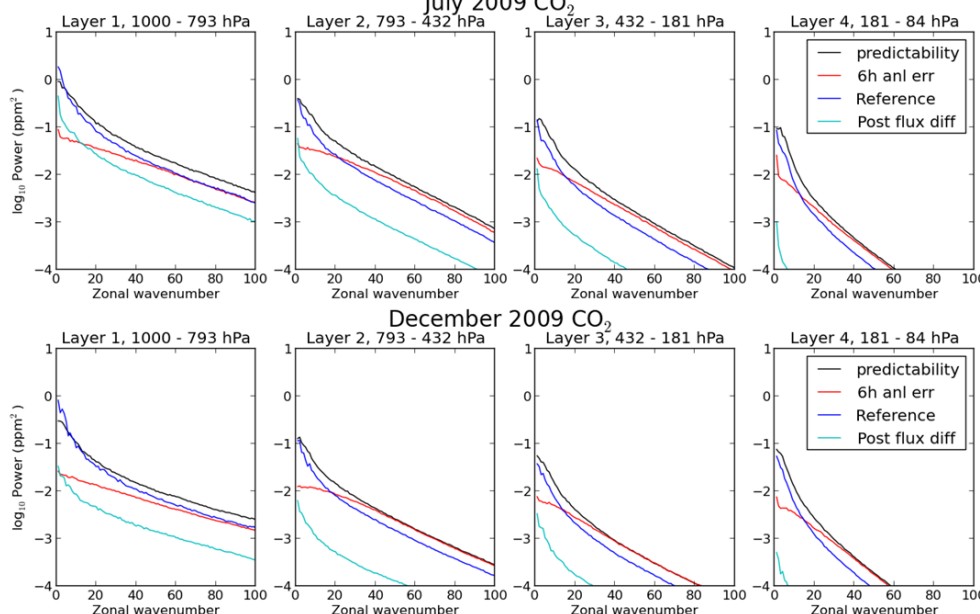

**Figure 12: As in Figure 11 but for spectra as a function of zonal wavenumber.**




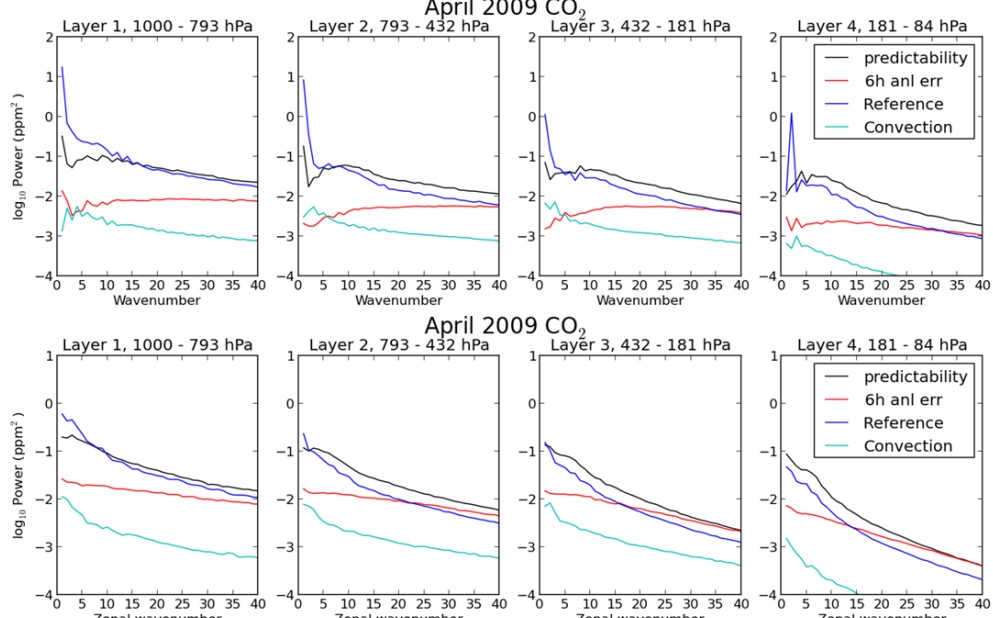

**Figure 13:** As in Figure 11 but for spectra in April 2009 and now errors due to the addition of convective tracer transport are shown in cyan curves. Spectra are shown as a function of total wavenumber (top row) or zonal wavenumber (bottom row). Note that only wavenumbers up to 40 are shown.