# Peer review of "Greenhouse gas simulations with a coupled meteorological and transport model: The predictability of CO2"

_Atmospheric Chemistry and Physics, 2016_

## Referee Comment (RC1) · Anonymous Referee #1 · 9 Jun 2016

This paper presents the adaptation of the GEMS-MACH model to simulate atmospheric CO$_2$ within an NWP framework. An evaluation of the simulations using different types of observations is performed. The question of predictability associated with different aspects of the model (growth of transport errors, imperfect initial conditions, and impact of convection) is also addressed. The paper is very well written and the predictability study shows very interesting results that are relevant to the atmospheric composition and carbon cycle community. So I would recommend the paper to be accepted subject to minor corrections. A list of general and more specific comments can be found below.
**GENERAL COMMENTS**

- The potential role of the $CO_2$ fluxes in the predictability of atmospheric $CO_2$ is not clearly presented. Although this is not the main focus of this study, the generally important role of the fluxes in the predictability of $CO_2$ should be emphasized. The relevance of the fluxes would become more evident if the predictability diagnostic would include a measure of the evolution of mean error with forecast lead time. In the paper two different surface fluxes are used but their differences are not explained. A description of the main differences (not only in formulation but in actual flux difference) would help the interpretation of the results in terms of flux errors. Can their difference be used to test the impact of flux errors on the predictability?

- The characterisation of analysis errors using a 6-hour shift seems an unrealistically large estimate of the analysis error. If that is the case, the results might not be indicative of the real impact of using imperfect analyses in $CO_2$ models (e.g. in flux inversions or in forward simulations), but will only provide an upper limit for scales that can be predicted even with the use of inaccurate initial conditions for the meteorology. I think it would be useful to compare this specification of uncertainty with the uncertainties provided by your analysis system.

- The comparison of the errors associated with imperfect analysis, the predictability and omission of convective transport as well as flux errors are done for 1 to 2 month-long simulations. It is not clear whether the same results would hold for shorter simulations (e.g. 5-day or 10-day simulations) when the predictability error is not saturated.

- Finally, it would be useful to have a clearer message of the implications of the findings regarding the impact of the different error sources for the carbon cycle community.

**SPECIFIC COMMENTS**

- The CAMS $CO_2$ analysis and forecasting system is using the Bermejo and Conde (2002) mass fixer since 2015 with the introduction of the high resolution $CO_2$ forecast. A paper has been recently submitted to GMD to document it:

  A. Agusti-Panareda, M. Diamantakis, V. Bayona, F. Klappenbach, and A. Butz (2016): Improving the inter-hemispheric gradient of total column atmospheric $CO_2$ and CH4 in simulations with the ECMWF semi-Lagrangian atmospheric global model (submitted to GMD)

- Page 11, line 15: Why should the the mass mixing ratio of $CO_2$ be adjusted to account for changes in the atmospheric pressure associated with analysis increments? Although they will affect the conservation of $CO_2$ mass in the model, the associated error is small and it does not grow with time. So why can't this error be considered part of the small error in mass conservation associated with model errors in atmospheric pressure?

- Page 13, line 14: How frequently are the retrieved fluxes from Carbon Tracker updated?

- Page 13, line 11: It is not clear what is the purpose of using two versions of the Carbon Tracker fluxes. What is the difference between CT2010 and CT2013B apart from the fact that the latter extends further ahead in time? A figure showing the difference in the seasonal cycle and the annual mean distribution might help to understand how large the differences are and where/when they are more pronounced.

- Page 14, lines 23-24: This line gives the impression that improving the diurnal cycle of $CO_2$ in the boundary layer is only a matter of improving the parameterisation of turbulent diffusion. Whereas the atmospheric $CO_2$ diurnal cycle near
the surface also depends on the diurnal cycle of the $CO_2$ fluxes. So improving the turbulence mixing parameterisation will not necessarily improve the $CO_2$ diurnal cycle, in particular if there are compensating errors in the retrieved fluxes. Also, since the retrieved fluxes are not using observations of the $CO_2$ diurnal cycle, their diurnal cycle will be mainly reflecting the prior fluxes. This means that the error in the atmospheric $CO_2$ diurnal cycle will have an even larger component from the $CO_2$ flux error.

• Page 14, lines 30-31: "The GEM-MACH-GHG simulations with different posterior fluxes are closer to each other than to CarbonTracker in the tropical Atlantic". I see large differences with CarbonTracker in the Arctic and Tropical Pacific too. Again, it would be useful to know the difference in fluxes between CT2010 and CT2013. If their difference is very small, then it is not surprising that the two simulations are so similar and we mainly see the differences between the CT transport and the GEMS-MACH transport.

• Page 15, lines 1-2: I do not understand how the finding that the fluxes are more important for explaining the tropical structure can be linked to the finding that posterior fluxes are sensitive to prior fluxes in the tropics.

• Page 15, lines 24-25: The main reason why the biases here are lower than those from Massart et al. (2016) is because of the use of retrieved fluxes. If the retrieved fluxes from flux inversion systems are well constrained by the observed annual growth rate (which is similar in all background stations) then they should produce annual biases close to zero with forward models. The GOSAT data assimilated by Massart et al. (2016) is too spare to be able to constrain the global growth rate by just adjusting the atmospheric concentrations using a 12-hour window.

• Page 15, lines 25-26: I don't agree with this hypothesis. The main difference between the results here and those by Massart et al. (2016) is the fact that the

retrieved fluxes are well constrained by the observed annual global atmospheric growth rate from surface observations, which should be similar to the growth rate observed in most TCCON stations. The system used by Massart et al (2016) is not adjusting the fluxes using long assimilation windows and therefore it is having a harder time constraining the annual global growth rate.

- Page 15, line 30: excluding Eureka, Karlsruhe and Izana.

- Page 16, line 5: The statement "has GEM-MACH-GHG exceptionally good agreement" is not appropriate unless it is specified that the agreement is better than CT in the free troposphere but all the models have similar error magnitude in the boundary layer.

- Page 16, line 8: "excellent annual results *in the free troposphere* because . . . "

- Page 16, lines 12-13: In autumn the gradient seems to be "too large". In the paper it says it is "too small". Moreover, the explanation that the vertical mixing just above the boundary layer is too weak would not make sense if the gradient was too small.

- Page 16, line 15: The departure of the vertical gradient from those observed can only be attributed solely to the model formulation if the two posterior fluxes used are significantly different. In the paper there is no evidence that this is the case for the profiles used over Canada.

- Page 16, lines 17-18: On a season by season basis, I don't think it is possible to say that "Overall, compare to Carbon Tracker, vertical gradients in GEM-MACH-GHG agree better with independent measurements in the mid troposphere but less well in the lower troposphere." This is only the case for the annual mean, but as mentioned in the paper the annual mean does not reflect the errors associated with the transport, but it reflects that fact that the errors have opposite sign in different seasons. So I would remove this statement.

- Page 16, line 23: what about the errors associated with the unresolved (parameterised) transport?

- Page 16, lines 25-26: The assumption of the fluxes being "perfectly known" is unrealistic.

- Page 17, line 5: The synoptic variability of the biogenic $CO_2$ fluxes can also have an impact on the synoptic variability of atmospheric $CO_2$ (Chan et al. 2004 in Tellus B, Agusti-Panareda et al. 2014 in ACP).

- Page 17, lines 8-10: It is important to also mention that the predicatibility error should include the random error as well as the bias. This paper addresses mainly the limits of predictability in terms of variance but often the more challenging problem for models using NWP assimilation windows is how to deal with the large-scale growing biases in the background air.

- Page 18: The number of days where the forecast is deemed to have skill is just an empirical measure that is useful in the context of comparing different parameters and different factors (as mentioned in the discussion section). I would emphasize that the numbers of 2-3 days and 5 days should not be taken as a theoretical limit (see line 27 in the paper). The reason why the predictability seems to be longer in the boundary layer could explained by the normalization of the predictability score, as well as the stronger influence from the prescibed fluxes. The normalisation of the random error will imply that the layers with larger zonal variability (e.g. near surface or in upper troposphere/lower stratosphere with well-defined zonally propagating waves) will have lower values of the predictability diagnostic. It is also worth noting that the specific predictability diagnostic used in this paper focuses on the ability of the model to simulate the expected zonal variability. However, there could be many other measures focusing on other aspects of the predicatibility (eg.anomaly correlations as done by Massart et al, 2014, or mean

error growth) that are not shown in this study. These should be made clarified in the paper.

- Page 20, line 3: Which surface fluxes?

- Page 20, lines 20-21: It is not clear how the spectra shown in Fig 11 is computed, specially the specification of the total and the zonal wave numbers shown in Fig 12 and 13.

- Page 20, lines 27-28: Does this mean that the spatial scales of the predictability errors are only assessed for a 1 to 2 month simulation?

- Page 21: Although the emphasis is on the predictability of the meterological parameters affecting the predictability of $CO_2$, the influence of the $CO_2$ fluxes in a real forecast setting (i.e. where the fluxes cannot be retrieved) should be emphasized. It seems as if they just play a secondary role as they are just mentioned as a remaining factor after all the others have been considered. A lot of emphasis is given to the role of the ocean and land surface conditions, but from Fig S7 I would say that there is still predictabtility at large scales even when those surface conditions are shifted by 3 months. I think this proves that at large-scales the fluxes are really the dominant factor for the predictability.

- Page 22, lines 1-2: I do not understand why the results are consistent with the transport biases acting at large-scales. Isn't the limit of predictability associated with imperfect analysis most relevant for small scales?

- Page 22, lines 13-16: I would say that the larger influence of the flux differences near the surface is not because the retrieved fluxes assimilated observerations near the surface, but because any surface flux will have a much larger influence on the $CO_2$ near the surface than at upper levels. The diminishing impact of flux differences with height has nothing to do with whether the observations assimilated were near the surface, in the mid troposphere or for the total column, but

on the fact that the influence of any surface flux will always diminish with height because of the transport and mixing away from the source/sink region.

- Page 22, lines 28-29: This sentence is not clear.

- Page 23, line 9: Can you associate the zonal spectra to the tropics exclusively? What about the Rossby wave in mid-latitudes?

- Page 23, lines 24-25: I think it would be useful to add the range of errors found on the seasonal time-scale when you say that the model compares well with observations(e.g. within +-2ppm based on Fig 8).

- Page 23, line 27: The statement that the gradient in the model is excellent can be misleading as it only applies to the annual mean. Since the gradient for specific seasons is what really reflects the transport uncertainties, I would say that the gradient is slightly overestimated in the free troposphere probably due to a lack of mixing in the model.

- Page 23, line 31: This is not the first time when the predictability of $CO_2$ has been investigated (see Massart et al. 2016, ACP). The results from Massart et al. (2016) show the forecast of column-averaged $CO_2$ has skill up to day 5, so the 2-3 day is not a theoretical limit, but it is dependend on the diagnostic used. This should be clarified somehow.

- Page 24, line 5: The climate predictability is important, but so is the predictability of the $CO_2$ fluxes.

- Page 24, line 6: If I understood well, the impact of imperfect analysis is tested by shifting the analysis field by 6 hours. This is not equivalent to "a 6 h forecast error" as mentioned in the paper.

[Figure]

**MINOR COMMENTS**

- Page 14, lines 1-2: It is difficult to see the blue line associated with the model in the middle panel.

- Figure 1: I think the sentences "..there should be 4 times as many blue boxes as shown. Some of these were omitted for clarify of presentation" could be misleading if the meteo analysis in the $CO_2$ model is only used at 00 UTC. If not, this should be clarified in the paper.

- Figures 3: It is difficult to see the blue line in the middle panel.

- Page 15, line 1 : replace "dynamics" by "transport".

---

## Referee Comment (RC2) · Anonymous Referee #2 · 15 Jun 2016

The paper includes two parts. It first describes a new coupled meteorological and tracer transport model based on Environment and Climate Change Canada's operational weather and prediction model, and then discusses the predictability of CO2 due to initial state sensitivity and land and ocean surface states. While the paper devotes most of the space to describe the steps adapting the GEM-MACC to do tracer transport, both the title and abstract do not reflect such effort. I would recommend dividing the current paper into two papers: one is on model development that is more suitable for Geophysical Model Development (GMD), and the other is on CO2 predictability, which may be suitable for ACP. At the current stage, the paper reads more like a model development paper. The following are my detailed comments:

[Figure]

1) The term of mass-conservation is loosely used. In CO2 transport, what needs to be conserved should be the total number of molecules in the atmosphere, not mixing ratio.

2) The reason of using 24-hour tracer-transport forecast cycle is not well illustrated in the text. Though spurious gravity wave could be generated during forecast, NWP has been using filtering technique to reduce spurious gravity waves (Nezlin et al., 2009). In the off-line tracer transport model, the meteorology analysis fields are read in every 6 hours, while this paper uses 24-hour. The paper only compares the 24-hour forecast to other reanalysis products. The accuracy of 24-hour meteorology forecast compared to the analysis fields read in by GEM-MACC needs to be assessed. Also, the sensitivity of CO2 transport to the length of tracer-transport forecast-cycle needs to be quantified, which can be addressed by changing the forecast cycle to 6 hours.

Nezlin, Y., S. Polavarapu, and Y. J. Rochon (2009), A new method of assessing filtering schemes in data assimilation systems, Q. J. R. Meteorol. Soc., 135, 1059–1070.

3) Figure 5 and 6 qualitatively compare the GEM CO2 fields to CarbonTracker. A figure showing the difference between GEM CO2 and CarbonTracker will be more quantitative. Also, it would be helpful to include a column CO2 north-south gradient comparison. Figure 5 indicates that CarbonTrakcer and GEM may have quite different N-S gradient.

4) The explanations for the disagreement between obs and aircraft and some surface insitu observations are not convincing. Figure 3 a and figure 8 b and c show that the summer draw down at lower levels simulated by the GEM model is much stronger than observations. The authors attribute this to the accuracy of underlying fluxes (P15, L33), since the model simulated CO2 agrees better with the observations when using the posterior fluxes constrained by GOSAT. While the accuracy of underlying fluxes could be the reason, I think it is more likely due to the accuracy of convective transport because the aircraft observations are over NA where surface flask observations

are dense. On the contrary, both figures seem to indicate that the vertical mixing at lower levels is too weak during both summer and winter. I suggest the authors using MACC III that is constrained by surface flask observations to do one more $CO_2$ forward simulations, and then compare GEM $CO_2$ to MACC III $CO_2$ fields.

5) In adapting the model to do $CO_2$ transport, the authors do not include the horizontal diffusion (section 2.3). What is the impact of horizontal diffusion on $CO_2$ fields? The authors at least can run an experiment including horizontal diffusion for few day and then compare to the control run. Within a few days, the mass conservation is not that critical.

6) In the second part of the paper, the authors try to quantify the $CO_2$ predictability in weather time scales and seasonal time scales. The discussions are lack of physical interpretation of the results and the implication of the results for flux inversion. Unlike meteorology fields and other air quality variables, such as O3, $CO_2$ transport and $CO_2$ observations are mainly used to quantify surface fluxes, not for prediction; the $CO_2$ prediction itself does not have any applications. Under this context, it is important to illustrate the connection between $CO_2$ predictability discussed here and the $CO_2$ flux inversions.

7) The authors simulate the analysis errors by shifting the met analysis by 6 hours. This would create large errors due to inaccurate description of diurnal cycle, which is especially significant over extratropics. Since ECCC has a hybrid approach to simulate background error covariance (page 10, line 24), the analysis error can be approximated by the spread of ensemble forecasts used in the hybrid scheme, which should be more realistic.

---

## Author Comment (AC1) · 20 Aug 2016

**Response to Review 1**

The impact of meteorological analysis uncertainties on the spatial scales resolvable in CO2 model simulations" by Saroja M. Polavarapu et al.

Original comments are in black text.  Our responses are in blue text.

**Unrequested modification**:  In addition to changes prompted by reviewers' questions, we decided to change the predictability metric from the square root of the global mean of zonal variance to a more intuitive one: global mean of zonal standard deviation.  In doing so, a coding error in the original diagnostic was found that affected only vorticity and divergence:  abnormally large normalized predictability error was produced in the stratosphere and mesosphere.  Since the square root and global mean operators do not commute, the change in Figures 9-10 resulting from the change in diagnostic did not have to be small, but were in fact minimal.  Thus the changes seen in the revised Figs. 9-10 are mainly from correcting the coding error.

This paper presents the adaptation of the GEMS-MACH model to simulate atmospheric  CO2 within an NWP framework. An evaluation of the simulations using different types of observations is performed. The question of predictability associated with different aspects of the model (growth of transport errors, imperfect initial conditions, and impact of convection) is also addressed. The paper is very well written and the predictability study shows very interesting results that are relevant to the atmospheric composition and carbon cycle community. So I would recommend the paper to be accepted subject to minor corrections. A list of general and more specific comments can be found below.

**Response:** We thank the Reviewer for the encouraging remarks and for the thorough reading of the paper. We believe that the revised manuscript has greatly benefitted from the comments of both Reviewers.

**GENERAL COMMENTS**
- The potential role of the CO2 fluxes in the predictability of atmospheric CO2 is not clearly presented. Although this is not the main focus of this study, the generally important role of the fluxes in the predictability of CO2 should be emphasized.  The relevance of the fluxes would become more evident if the predictability diagnostic would include a measure of the evolution of mean error with forecast lead time. In the paper two different surface fluxes are used but their differences are not explained. A description of the main differences (not only in formulation but in actual flux difference) would help the interpretation of the results in terms of flux errors. Can their difference be used to test the impact of flux errors on the predictability?
  **Response:**  As the reviewer notes, the role of $CO_2$ fluxes in the predictability of atmospheric $CO_2$ is not a main focus here and this is partly because the only fluxes we had available at the time were two sets of retrieved fluxes from CarbonTracker.  As implied by the Reviewer, the use of two essentially equivalent retrieved fluxes does not shed light on the impact of flux errors on predictability.  More can be learned by comparing prior and posterior fluxes but we did not have access to this.  As noted in the discussion section of the original manuscript (p.24, lines 27-35), some interesting extensions and applications to different observing systems can be done and are now in progress.  The only reason that these two sets of fluxes were used was to illustrate this potential.  However, it is clear that this potential is not being well illustrated as hoped.  Moreover, it seems to be detracting from the main message.  Therefore, we have removed all references to the two sets of fluxes and used only CT2013B which is the preferred product (according to Andy Jacobson of NOAA).  In addition, we have tried to emphasize the important role of fluxes in

predictability results throughout the text of the revised manuscript. As for bias, we have computed the global mean of predictability error normalized by the standard deviation (as in Fig. 9 of the revised manuscript) and included it below. For $CO_2$, the bias is generally negligible compared to the standard deviation in the troposphere. This is likely because in our experimental design, the flux of the reference and perturbed cycle is the same. Moreover, model errors and initial $CO_2$ state errors are also eliminated because we focus on meteorological state errors. Bias is quite prevalent however in model forecast to measurement comparisons where flux, initial state, meteorology, model formulation, observation and representativeness errors are present. A paragraph discussing the important of forecast bias was added to the discussion section of the revised manuscript.

[Figure]

**Figure R1-1:** Global mean difference between the reference and climate-mode cycles during January 2009. Values are normalized by the global mean zonal standard deviation of the initial state. Mean departures are negligible for the wind field (relative vorticity and divergence fields), are important for the temperature near the tropopause and negligible for $CO_2$ in the troposphere.

- The characterisation of analysis errors using a 6-hour shift seems an unrealistically large estimate of the analysis error. If that is the case, the results might not be indicative of the real impact of using imperfect analyses in CO2 models (e.g. in flux inversions or in forward simulations), but will only provide an upper limit for scales that can be predicted even with the use of inaccurate initial conditions for the meteorology. I think it would be useful to compare this specification of uncertainty with the uncertainties provided by your analysis system.

**Response:** The reviewer is correct that a 6h shift could be large compared to a 6h analysis error. But as explained in the manuscript, this was done because we did not have access to analysis errors in 2009-2010. While an ensemble Kalman Filter has been operational at ECCC for some time, results were not archived. Archives begin in 2011 but are not reliable. Looking at EnKF ensemble spread estimates for January 2012 (a different time period), we see that analysis errors are indeed smaller than 6h difference errors but there is no simple pattern to the scaling factor—it depends on time, location, height and variable. Not surprisingly, no proxy can replace a true analysis error even though the analysis error is itself an imperfect estimate. Of course, as noted on p25, lines 12-15, we are in the process of developing an EnKF and will someday have access to actual analysis errors from our own system. When that system is available we will be able to quantify the limiting spatial scales due to analysis uncertainties. For the purpose of this work, we agree that it is important to emphasize the fact that the 6h shift is an overestimate of analysis errors and this is done in the revised manuscript. But the main conclusions regarding uncertain analyses, namely, that (1) some scales in $CO_2$ cannot be resolved due to the presence of analysis uncertainties, and (2) that for small enough spatial scales, the error spectrum due to analysis uncertainties asymptotes to the predictability spectrum hold regardless of the size of the error.

- The comparison of the errors associated with imperfect analysis, the predictability and omission of convective transport as well as flux errors are done for 1 to 2 month-long simulations. It is not clear whether the same results would hold for shorter simulations (e.g. 5-day or 10-day simulations) when the predictability error is not saturated.
  **Response:** Actually the simulations were done for one or two years but the spectra were averaged over a month in order to get more robust spectra. On short time scales like 5-10 days one would need an ensemble of short forecasts that can be averaged and we do not have that capability yet. This is routinely done at operational centres and when our assimilation cycle (rather than just forward model simulations) based on the ensemble Kalman filter is developed, this could be done. Here our focus is on the saturated error levels where robust statistics can be obtained by averaging spectra. Animations of daily spectra reveal that spectra are generally slowly varying in time and height but small noise exists so that averaging in time and height is appropriate. Our spectra are computed from roughly 31x12=372 samples. The question of short time scales for model and other components of transport error is, however, an interesting one that is left for future work with our ensemble Kalman filter once it is developed.

- Finally, it would be useful to have a clearer message of the implications of the findings regarding the impact of the different error sources for the carbon cycle community.
  **Response:** We agree that implications of our results for the carbon cycle community need to be much better presented. Therefore, we have completely rewritten the abstract and the discussion section, and revised the introduction. We have also now added a new section, just after the Introduction, which clarifies the relevance of predictability for flux inversions. In this section the terminology used in atmospheric dynamics and data assimilation is connected to that used in flux inversions. Prediction or forecast error is related to transport error, and we define the components of transport error, namely, flux error, initial $CO_2$ state error, meteorological state errors, and model error. With these definitions, the results and their implications are easier to explain.

**SPECIFIC COMMENTS**

- The CAMS CO2 analysis and forecasting system is using the Bermejo and Conde (2002) mass fixer since 2015 with the introduction of the high resolution CO2 forecast. A paper has been recently submitted to GMD to document it:
  *Agusti-Panareda, M. Diamantakis, V. Bayona, F. Klappenbach, and A. Butz (2016): Improving the inter-hemispheric gradient of total column atmospheric CO2 and CH4 in simulations with the ECMWF semi-Lagrangian atmospheric global model (submitted to GMD)*
  **Response:** Thanks for the update. The sentence on page 7 lines 1-2 has been changed to:

"For greenhouse gas transport, the ECMWF forecast model also uses the Bermejo-Conde scheme (Agusti-Panareda et al., 2016). "

- Page 11, line 15: Why should the the mass mixing ratio of CO2 be adjusted to account for changes in the atmospheric pressure associated with analysis increments? Although they will affect the conservation of CO2 mass in the model, the associated error is small and it does not grow with time. So why can't this error be considered part of the small error in mass conservation associated with model errors in atmospheric pressure?
  **Response:** The reviewer is correct that it is not necessary to account for this error since it is small and does not grow with time. We chose to account for it because it removes a noisy component of the mass budget plots and makes it easier to spot problems.

- Page 13, line 14: How frequently are the retrieved fluxes from Carbon Tracker updated?
  **Response:** Fluxes are provided every 3h and inserted every model time step (15 minutes). This information was added to the revised section 4.2.

- Page 13, line 11: It is not clear what is the purpose of using two versions of the Carbon Tracker fluxes. What is the difference between CT2010 and CT2013B apart from the fact that the latter extends further ahead in time? A figure showing the difference in the seasonal cycle and the annual mean distribution might help to understand how large the differences are and where/when they are more pronounced.
  **Response:** The two products are largely comparable. CT2013B is a later product and is thus viewed as "better" by NOAA. The only reason that both products were used is to see the impact of flux differences on spatial scales resolved. But because the two products are effectively equivalent, this point is not being well illustrated. Moreover, it is clear that the use of two equivalent flux products is detracting from the main conclusions regarding predictability. Therefore, we have removed the experiment using CT2010 from the manuscript (Figs. 4,5,6,8,11,12) in order to remove this distraction.

- Page 14, lines 23-24: This line gives the impression that improving the diurnal cycle of CO2 in the boundary layer is only a matter of improving the parameterisation of turbulent diffusion. Whereas the atmospheric CO2 diurnal cycle near the surface also depends on the diurnal cycle of the CO2 fluxes. So improving the turbulence mixing parameterisation will not necessarily improve the CO2 diurnal cycle, in particular if there are compensating errors in the retrieved fluxes. Also, since the retrieved fluxes are not using observations of the CO2 diurnal cycle, their diurnal cycle will be mainly reflecting the prior fluxes. This means that the error in the atmospheric CO2 diurnal cycle will have an even larger component from the CO2 flux error.
  **Response:** We agree with this point as it was not our intention to oversimplify the issue of boundary layer modelling. Therefore we have removed the last 3 sentences of this paragraph.

- Page 14, lines 30-31: "The GEM-MACH-GHG simulations with different posterior fluxes are closer to each other than to CarbonTracker in the tropical Atlantic". I see large differences with CarbonTracker in the Arctic and Tropical Pacific too. Again, it would be useful to know the difference in fluxes between CT2010 and CT2013. If their difference is very small, then it is not surprising that the two simulations are so similar and we mainly see the differences between the CT transport and the GEMS-MACH transport.
  **Response:** As noted above, it is clear that the use of two essentially equivalent sets of retrieved fluxes is not helpful, so the simulation with CT2010 was removed from this article.

- Page 15, lines 1-2: I do not understand how the finding that the fluxes are more important for explaining the tropical structure can be linked to the finding that posterior fluxes are sensitive to prior fluxes in the tropics.
  **Response:** With the removal of the GEM simulation with CT2010 fluxes, this sentence becomes irrelevant and was removed.

- Page 15, lines 24-25: The main reason why the biases here are lower than those from Massart et al. (2016) is because of the use of retrieved fluxes. If the retrieved fluxes from flux inversion systems are well constrained by the observed annual growth rate (which is similar in all background stations) then they should produce annual biases close to zero with forward models. The GOSAT data assimilated by Massart et al. (2016) is too spare to be able to constrain the global growth rate by just adjusting the atmospheric concentrations using a 12-hour window.
  **Response:** This is a good point. See response to the next point.

- Page 15, lines 25-26: I don't agree with this hypothesis. The main difference between the results here and those by Massart et al. (2016) is the fact that the retrieved fluxes are well constrained by the observed annual global atmospheric growth rate from surface observations, which should be similar to the growth rate observed in most TCCON stations. The system used by Massart et al (2016) is not adjusting the fluxes using long assimilation windows and therefore it is having a harder time constraining the annual global growth rate.
  **Response:** This argument makes sense. The sentence: "Since the analyses in Massart were based on GOSAT whereas CT2013B used surface data, this may indicate a better agreement of surface observations with TCCON." was changed to: "This is because the surface observations assimilated by CarbonTracker with long assimilation windows are able to constrain the global atmospheric growth rate whereas the system used by Massart et al. (2016) does not use long assimilation windows and thus does not constrain the global growth rate as effectively."

- Page 15, line 30: excluding Eureka, Karlsruhe and Izana.
  **Response:** Like Eureka, Karlsruhe and Izana also have large RMS, but this statement specifically discusses bias.

- Page 16, line 5: The statement "has GEM-MACH-GHG exceptionally good agreement" is not appropriate unless it is specified that the agreement is better than CT in the free troposphere but all the models have similar error magnitude in the boundary layer.
  **Response:** Agreed. The statement was modified to: "In the annual average, GEM-MACH-GHG has good agreement with these independent measurements while CT2013B has a very slight positive bias."

- Page 16, line 8: "excellent annual results *in the free troposphere* because . . . "
  **Response:** Agreed. The statement was modified as suggested.

- Page 16, lines 12-13: In autumn the gradient seems to be "too large". In the paper it says it is "too small". Moreover, the explanation that the vertical mixing just above the boundary layer is too weak would not make sense if the gradient was too small.
  **Response:** Thanks for catching this error. The statement was corrected.

- Page 16, line 15: The departure of the vertical gradient from those observed can only be attributed solely to the model formulation if the two posterior fluxes used are significantly different. In the paper there is no evidence that this is the case for the profiles used over Canada.

**Response:** This point is valid. Additionally, for reasons already mentioned, all references to GEM-MACH-GHG simulations with the second posterior flux were removed. So, this statement has been removed.

- Page 16, lines 17-18: On a season by season basis, I don't think it is possible to say that "Overall, compare to Carbon Tracker, vertical gradients in GEM-MACHGHG agree better with independent measurements in the mid troposphere but less well in the lower troposphere." This is only the case for the annual mean, but as mentioned in the paper the annual mean does not reflect the errors associated with the transport, but it reflects that fact that the errors have opposite sign in different seasons. So I would remove this statement.
  **Response:** Done.

- Page 16, line 23: what about the errors associated with the unresolved (parameterised) transport?
  **Response:** Transport here refers to the model's determinist transport including parameterized subgrid scale processes which are represented deterministically.

- Page 16, lines 25-26: The assumption of the fluxes being "perfectly known" is unrealistic.
  **Response:** We changed the sentence: "In a data assimilation or flux inversion system, when the fluxes are well constrained by observations, the ability to estimate the CO2 fields using observations will ultimately be limited by the loss of meteorological predictability, just as the quality of weather forecasting products are."

- Page 17, line 5: The synoptic variability of the biogenic CO2 fluxes can also have an impact on the synoptic variability of atmospheric CO2 (Chan et al. 2004 in Tellus B, Agusti-Panareda et al. 2014 in ACP).
  **Response:** Good point. The statement was modified to "The predictability problem on weather timescales is related to forecast sensitivity to initial conditions, but the atmospheric variability of $CO_2$ on diurnal (Law et al., 2008), synoptic (Chan et al. 2004, Agusti-Panareda et al. 2014) and seasonal and interannual (Gurney et al., 2002, 2004; Baker et al. 2006a, LeQuéré et al., 2015) time scales is largely governed by the terrestrial biospheric fluxes and hence is determined by sensitivity to boundary rather than initial conditions."

- Page 17, lines 8-10: It is important to also mention that the predicatibility error should include the random error as well as the bias. This paper addresses mainly the limits of predictability in terms of variance but often the more challenging problem for models using NWP assimilation windows is how to deal with the large-scale growing biases in the background air.
  **Response:** The bias is most often revealed when comparing forecasts to observations. In this case, forecast or transport model error is affected by flux errors convolved with meteorological state errors, initial $CO_2$ state errors and model formulation errors. Additionally, observation and representativeness errors are present in this comparison, but the most likely source of the bias is the convolution of flux and atmospheric transport. In our work, with a reference simulation, model formulation errors are absent. Flux and initial $CO_2$ state errors are also absent when we focus on the contribution of meteorological state errors to forecast (transport model) errors. Thus as shown in Figure R1-1, bias is not important in our study, but the Reviewer is correct that in the usual transport error assessments, it is an important concern. This point is made in a new paragraph in the revised discussion section.

- Page 18: The number of days where the forecast is deemed to have skill is just an empirical measure that is useful in the context of comparing different parameters and different factors (as mentioned in the discussion section). I would emphasize that the numbers of 2-3 days and 5 days

should not be taken as a theoretical limit (see line 27 in the paper). The reason why the predictability seems to be longer in the boundary layer could explained by the normalization of the predictability score, as well as the stronger influence from the prescibed fluxes. The normalisation of the random error will imply that the layers with larger zonal variability (e.g. near surface or in upper troposphere/lower stratosphere with well-defined zonally propagating waves) will have lower values of the predictability diagnostic. It is also worth noting that the specific predictability diagnostic used in this paper focuses on the ability of the model to simulate the expected zonal variability. However, there could be many other measures focusing on other aspects of the predicatibility (eg.anomaly correlations as done by Massart et al, 2014, or mean error growth) that are not shown in this study. These should be made clarified in the paper.

**Response:** Agreed. We tried to caution against viewing the numbers of 2-3 or 5 days as the result given their dependence on some arbitrary parameter choices made, in general. But we agree that the statement (p. 19, line 27) should be qualified. We also agree that the normalization may be playing a role in the relative predictability between layers, and this is now mentioned in the revised manuscript. In addition, the anomaly correlation results of Massart et al. (2015) are now discussed in the revised discussion section.

- Page 20, line 3: Which surface fluxes?
  **Response:** Those from the terrestrial biosphere. The text has been corrected.

- Page 20, lines 20-21: It is not clear how the spectra shown in Fig 11 is computed, specially the specification of the total and the zonal wave numbers shown in Fig 12 and 13.
  **Response:** As our model grid is defined on Gaussian latitudes, fields can be decomposed level by level into spherical harmonics following Boer (1983). Then using a triangular truncation and summing over either total or zonal wavenumber, the spectra are obtained as function of the remaining wavenumbers.

- Page 20, lines 27-28: Does this mean that the spatial scales of the predictability errors are only assessed for a 1 to 2 month simulation?
  **Response:** No, the simulation was for 1 year, the statistics were averaged over 1 month. By averaging the statistics, some of the noisiness of the spectra (seen in animations of daily spectra) is removed. One month was found sufficient to smooth the spectra yet allow for some seasonal variation to be seen.

- Page 21: Although the emphasis is on the predictability of the meterological parameters affecting the predictability of CO2, the influence of the CO2 fluxes in a real forecast setting (i.e. where the fluxes cannot be retrieved) should be emphasized. It seems as if they just play a secondary role as they are just mentioned as a remaining factor after all the others have been considered. A lot of emphasis is given to the role of the ocean and land surface conditions, but from Fig S7 I would say that there is still predictabtility at large scales even when those surface conditions are shifted by 3 months. I think this proves that at large-scales the fluxes are really the dominant factor for the predictability.
  **Response:** We agree. There was no intention to downplay the important role of $CO_2$ fluxes in predictability at large scales. The fact that the land and ocean surface play any role at all is rather interesting and novel (we believe). This is what we were trying to point out. However, we have modified the text (sections 5.2 and 6) to emphasize that we also demonstrate the dominant role of $CO_2$ fluxes on predictability at large scales.

- Page 22, lines 1-2: I do not understand why the results are consistent with the transport biases acting at large-scales. Isn't the limit of predictability associated with imperfect analysis most relevant for small scales?
  **Response:** Yes. It is the small scales that are the least predictable. But these unpredictable scales get surprisingly large as altitude increases. Chevallier et al. (2010) shows that small transport errors can produce flux biases of large spatial scale for GOSAT OSSEs. However, the connection is not that strong, so this statement was dropped.

- Page 22, lines 13-16: I would say that the larger influence of the flux differences near the surface is not because the retrieved fluxes assimilated observerations near the surface, but because any surface flux will have a much larger influence on the CO2 near the surface than at upper levels. The diminishing impact of flux differences with height has nothing to do with whether the observations assimilated were near the surface, in the mid troposphere or for the total column, but on the fact that the influence of any surface flux will always diminish with height because of the transport and mixing away from the source/sink region.
  **Response:** Quite true. However, this paragraph was removed in order to remove the distraction of having used two different but comparable sets of retrieved fluxes.

- Page 22, lines 28-29: This sentence is not clear.
  **Response:** The sentence was modified to: "*However, since predictability was already close to lost even with the correct land and ocean surface (power in predictability error is same order of magnitude as that for reference state), this conclusion may not be warranted the worsened predictability in July to September with incorrect land and ocean surface fields is not a clear indication of influence of their influence in the tropics.*"

- Page 23, line 9: Can you associate the zonal spectra to the tropics exclusively? What about the Rossby wave in mid-latitudes?
  **Response:** No, the zonal spectra are not exclusively related to the tropics, but the tropics tend to be better represented in zonal spectra. These diagnostics can only be taken as indications of tropical predictability, since the measure is not specific to the tropics. However, the statement concerned the experiment with two posterior fluxes so it was removed.

- Page 23, lines 24-25: I think it would be useful to add the range of errors found on the seasonal time-scale when you say that the model compares well with observations(e.g. within +-2ppm based on Fig 8).
  **Response:** Good idea. The statement was modified.

- Page 23, line 27: The statement that the gradient in the model is excellent can be misleading as it only applies to the annual mean. Since the gradient for specific seasons is what really reflects the transport uncertainties, I would say that the gradient is slightly overestimated in the free troposphere probably due to a lack of mixing in the model.
  **Response:** "excellent" was changed to "slightly overestimated"

- Page 23, line 31: This is not the first time when the predictability of CO2 has been investigated (see Massart et al. 2016, ACP). The results from Massart et al. (2016) show the forecast of column-averaged CO2 has skill up to day 5, so the 2-3 day is not a theoretical limit, but it is dependend on the diagnostic used. This should be clarified somehow.
  **Response:** Agreed. A discussion of the results obtained by Massart et al. (2016) was added to the completely revised new discussion section.

- Page 24, line 5: The climate predictability is important, but so is the predictability of the CO2 fluxes.
  **Response:** Agreed. This whole section was rewritten. In doing so, the importance of the $CO_2$ fluxes on predictability results is better emphasized.

- Page 24, line 6: If I understood well, the impact of imperfect analysis is tested by shifting the analysis field by 6 hours. This is not equivalent to "a 6 h forecast error" as mentioned in the paper.
  **Response:** True. We have taken the 6h shift in analyses as a proxy for 6 h forecast error, but it should not be identified as such. This statement was clarified to avoid making the equivalence. Additionally, we have reviewed the entire manuscript and eliminated other instances where 6h analysis error is mentioned when referring to the proxy.

**MINOR COMMENTS**
- Page 14, lines 1-2: It is difficult to see the blue line associated with the model in the middle panel.
  **Response:** We agree. We had tried to reorder the colours but one of the curves will always be hard to see. As it is, this panel shows that the seasonal variation of the diurnal cycle amplitude is captured well. Where the blue curve is difficult to see is in the summer, and a better way to assess this is by computing and comparing the diurnal cycle amplitude to observations. This is done in Figure 4. We did however modify this plot to reduce the obscuring of the curves by the legend in the bottom panel.

- Figure 1: I think the sentences "..there should be 4 times as many blue boxes as shown. Some of these were omitted for clarify of presentation" could be misleading if the meteo analysis in the CO2 model is only used at 00 UTC. If not, this should be clarified in the paper.
  **Response:** The meteorological analyses were only used at 00 UTC although the CMC produces them every 6 h. These sentences were removed from the caption.

- Figures 3: It is difficult to see the blue line in the middle panel.
  **Response:** This is the same point mentioned two bullets ago.

- Page 15, line 1: replace "dynamics" by "transport".
  **Response:** Done.

---

## Author Comment (AC2) · 20 Aug 2016

**Response to Review 2**

The impact of meteorological analysis uncertainties on the spatial scales resolvable in CO2 model simulations" by Saroja M. Polavarapu et al.

Original comments are in black text. Our responses are in blue text.

**Unrequested modification**: In addition to changes prompted by reviewers' questions, we decided to change the predictability metric from the square root of the global mean of zonal variance to a more intuitive one: global mean of zonal standard deviation. In doing so, a coding error in the original diagnostic was found that affected only vorticity and divergence: abnormally large normalized predictability error was produced in the stratosphere and mesosphere. Since the square root and global mean operators do not commute, the change in Figures 9-10 resulting from the change in diagnostic did not have to be small, but were in fact minimal. Thus the changes seen in the revised Figs. 9-10 are mainly from correcting the coding error.

We thank the Reviewer for the comments. We believe that the revised manuscript has greatly benefitted from the comments of both Reviewers.

The paper includes two parts. It first describes a new coupled meteorological and tracer transport model based on Environment and Climate Change Canada's operational weather and prediction model, and then discusses the predictability of $CO_2$ due to initial state sensitivity and land and ocean surface states. While the paper devotes most of the space to describe the steps adapting the GEM-MACC to do tracer transport, both the title and abstract do not reflect such effort. I would recommend dividing the current paper into two papers: one is on model development that is more suitable for Geophysical Model Development (GMD), and the other is on $CO_2$ predictability, which may be suitable for ACP. At the current stage, the paper reads more like a model development paper.

**Response:** Before starting this manuscript we had considered writing a separate paper just documenting the model. However, we felt that although it was an arduous, painstaking task to adapt GEM-MACH to simulate $CO_2$, this work alone would not justify publication. This is because of the intrinsic important of fluxes to $CO_2$ simulations. Since we do not yet estimate $CO_2$ fluxes, only the transport component of simulations can be assessed, and only to a limited degree since using retrieved fluxes from another model means that model's transport errors are convolved with ours. This is an important point that is demonstrated here. It makes sense to document a system which is semi-operational, near-real time, or which estimates fluxes. On the other hand, using a coupled meteorology and transport model has some advantages that we explore here, namely, the spatial scales on which meteorological uncertainties impact transport error. These are highly original and significant results relevant to the understanding of $CO_2$ assimilation results. This is what is described in the main portion of the paper. However, this being the first instance of the use of this model for greenhouse gas simulation, it was necessary to document the model components and show a few results to assure the reader that predictability results, which are necessarily model dependent, and shown in section 4 are generally relevant. However, we agree with the Reviewer that this was not well indicated by the title and abstract of the original manuscript. Accordingly, we have modified the title in the revised manuscript to: *Greenhouse gas simulations with a coupled meteorological and transport model: The predictability of carbon dioxide*. Moreover, the abstract, Introduction, and discussion have been revised to obtain a more coherent picture focused on the coupling of $CO_2$ and weather and climate. A new section was also added to connect the ideas of predictability and transport model error. Since the aspect of transport model error that is difficult to

address with an offline model is the error due to uncertain meteorology, the use of a coupled meteorological and transport model is critical for this work.

The following are my detailed comments:

1) The term of mass-conservation is loosely used. In $CO_2$ transport, what needs to be conserved should be the total number of molecules in the atmosphere, not mixing ratio.

   **Response:** The mass budget of $CO_2$ has been a primary objective in the development of our modelling system. We spent a significant amount of time to make changes to the NWP model to ensure the conservation of the total amount of $CO_2$ molecules during the full length of the simulation. In section 2 we describe changes made to the model to express $CO_2$ in terms of dry mixing ratio and the effort made to conserve dry air in the model. Nevertheless we have revised the text to avoid any confusion. In section 2.2 the text has been modified to explicitly mention how we compute global mass from mixing ratio. In section 2.3, the line mentioning the conservation of mixing ratio has been removed. That statement had existed only to explain why we do not use horizontal diffusion, but it was removed to avoid any possible confusion.

2) The reason of using 24-hour tracer-transport forecast cycle is not well illustrated in the text. Though spurious gravity wave could be generated during forecast, NWP has been using filtering technique to reduce spurious gravity waves (Nezlin et al., 2009). In the off-line tracer transport model, the meteorology analysis fields are read in every 6 hours, while this paper uses 24-hour. The paper only compares the 24-hour forecast to other reanalysis products. The accuracy of 24-hour meteorology forecast compared to the analysis fields read in by GEM-MACC needs to be assessed. Also, the sensitivity of $CO_2$ transport to the length of tracer-transport forecast-cycle needs to be quantified, which can be addressed by changing the forecast cycle to 6 hours. Nezlin, Y., S. Polavarapu, and Y. J. Rochon (2009), A new method of assessing filtering schemes in data assimilation systems, Q. J. R. Meteorol. Soc., 135, 1059–1070.

   **Response:** Our experiments have shown that the quality of the $CO_2$ simulations does not necessarily improve with more frequent cycling. Figure R-2.1 below (not shown in the manuscript) indicates indeed that the spatial distribution of $CO_2$ is not sensitive to the cycling frequency. By doing 24 hr cycling we reduce significantly the amount of CPU without sacrificing anything in terms of quality of results. The actual impact on the $CO_2$ evolution was not large in terms of $XCO_2$ (Fig. R2-1). The differences seem random in both space and time (from animations of figures like Fig. R2-1). However, differences do accumulate in the stratosphere and mesosphere presumably due to different wave generation characteristics which impact the slow overturning Brewer-Dobson circulation. The 24h update yielded better (slower) transport in the middle atmosphere (for reasons not yet known). Although the mass of $CO_2$ in the stratosphere and mesosphere is not important, this difference in circulation meant that the 24h update cycle was preferred. As for filtering techniques, this was a concern when developing our system's configuration. However, we found that it had very little impact with a 24h cycle and was more expensive to run.

[Figure]

Fig. R2-1: The CO$_2$ state on July 1, 2009 00Z from 2 model simulations which are identical except for a 6h update cycle (top) and a 24h update cycle (middle) and differences between the top and middle panels (bottom). Difference patterns change daily but remain on fine spatial scales.

For this work, it is important to demonstrate that our 24h forecasts are within the uncertainty of reanalyses. As the base model is an operational model, its forecasts are routinely verified against measurements and compared with other operational centres (as noted in the original manuscript). But what is more relevant to the carbon cycle community, and to this paper, is to demonstrate that stitching together a sequence of 24h forecasts has relevance to a time series of reanalyses since the latter can be used to constrain flux inversions. Therefore we could compare our analysis errors to that of reanalyses. However, we went much further than that and demonstrated that the entire forecast (up to 24h) is as close to a reanalysis as are any two reanalyses datasets. The point is that reanalyses are not perfect and the difference in reanalysis datasets is a measure of their uncertainty (and is commonly used as such in the climate science community). Thus our full time series of disjoint 24h forecasts has no more uncertainty than the time series of reanalyses. This point is now explained in the last paragraph of section 4.1 of the revised manuscript.

3)  Figure 5 and 6 qualitatively compare the GEM CO$_2$ fields to CarbonTracker. A figure showing the difference between GEM CO$_2$ and CarbonTracker will be more quantitative. Also, it would be helpful to include a column CO$_2$ north-south gradient comparison. Figure 5 indicates that CarbonTrakcer and GEM may have quite different N-S gradient.

`

**Response:** Figures 5 and 6 have been replaced so that now one panel depicts the difference between GEM and CarbonTracker. The text has been correspondingly adjusted.

4) The explanations for the disagreement between obs and aircraft and some surface insitu observations are not convincing. Figure 3 a and figure 8 b and c show that the summer draw down at lower levels simulated by the GEM model is much stronger than observations. The authors attribute this to the accuracy of underlying fluxes (P15, L33), since the model simulated $CO_2$ agrees better with the observations when using the posterior fluxes constrained by GOSAT. While the accuracy of underlying fluxes could be the reason, I think it is more likely due to the accuracy of convective transport because the aircraft observations are over NA where surface flask observations are dense. On the contrary, both figures seem to indicate that the vertical mixing at lower levels is too weak during both summer and winter. I suggest the authors using MACC III that is constrained by surface flask observations to do one more $CO_2$ forward simulations, and then compare GEM $CO_2$ to MACC III $CO_2$ fields.

**Response:** In Fig. 3a the disagreement between the model and Alert observations was attributed to fluxes because different retrieved fluxes from GEOS-Chem (using GOSAT) observations did not have this mismatch. Thus, we can conclude a mismatch in long range transport between CarbonTracker and GEM-MACH-GHG. However, as noted by the Reviewer, the problem with the GEOS-Chem flux integration is that the observing system also changed along with the source of fluxes. While using MACC III posterior fluxes constrained by surface observations would help settle the issue, we also now have access to a similar dataset, namely, GEOS-Chem posterior fluxes constrained by only surface observations, as well as other new information. This additional information supports our original point of view. Firstly, a GEM-MACH-GHG simulation with GEOS-Chem fluxes constrained only by surface observations also agrees better with observations than when CarbonTracker fluxes are used (Fig. R2-2). Secondly, GEM-MACH-GHG constrained with CarbonTracker-CH4 fluxes also agrees well with measurements at Alert. Since the transport is identical for both $CO_2$ and $CH_4$ (both were carried in the same model) yet only $CO_2$ has a disagreement with surface observations at high latitudes (Fig. R2-3), the CarbonTracker $CO_2$ fluxes are clearly implicated. That is, the CarbonTracker transport errors do not match GEM-MACH-GHG transport errors in high latitudes in the autumn. With this additional information, we can now safely attribute the mismatch at high latitudes to the fluxes and hence to the mismatch of GEM and CarbonTracker transport at high latitudes. Figure S6 of the supplemental material was modified to include Fig. R2-2 and text was correspondingly modified. However, we did not add Fig. R2-3 to the supplemental material because our $CH_4$ simulations were not described in this article. This will be the topic of a forthcoming article (not yet in preparation). As for convective transport, our updraft velocities compare well to observed values, but we do not yet have tracer transport through the shallow convection scheme, and this could explain the overestimated vertical gradient seen in Figure 8. The shallow convection scheme is now described in section 3.5 and this point is now mentioned in connection with Fig. 8.

[Figure]

Figure R2-2: Time series of $CO_2$ at Alert. GEM-MACH-GHG simulations with GEOS-Chem posterior fluxes obtained with GOSAT (red) and with surface observations only (blue) are compared to

measurements (black) for July 2009-Dec 2010.  Note the agreement with observations in autumn for both model simulations.

[Figure]

Figure R2-3:  Time series of $CH_4$ (top) and $CO_2$ (bottom) at Alert.  A recent GEM-MACH-GHG run with both $CO_2$ and $CH_4$ using posterior fluxes from CT2013B (for $CO_2$) and CarbonTracker-CH4 (for $CH_4$) (blue) is compared to measurements (black).  Note that there is no disagreement in autumn for $CH_4$ as there is for $CO_2$ with the same model transport.

5) In adapting the model to do $CO_2$ transport, the authors do not include the horizontal diffusion (section 2.3). What is the impact of horizontal diffusion on $CO_2$ fields? The authors at least can run an experiment including horizontal diffusion for few day and then compare to the control run. Within a few days, the mass conservation is not that critical.

**Response:**  During the model development (a couple of years ago), we did in fact look at the impact of horizontal diffusion.  Not surprisingly, the $CO_2$ field is slightly smoother (the diffusion uses a del-6 operator) but there is an impact on global mass conservation on the time scale of one year.  As noted in the original manuscript (p.8, lines 13-14), given our desire for exact mass conservation, horizontal diffusion cannot be included.  Nevertheless, for interest sake, we show here a 2-year-old plot of spectra from a model run with the standard high order (del-6 operator) (red), a comparable run with no horizontal diffusion (black) and another with greatly enhanced diffusion (del-4 operator) (blue).  The impact of horizontal diffusion is on wavenumbers 100 and higher.  Similar results are seen for other vertical levels and other dates.

[Figure]

Figure R2-4:  Spectra of $CO_2$ as a function of zonal wavenumber for various model levels whose approximate pressure is given above each panel.  The black curves correspond to the control cycle while the red curves add horizontal diffusion with high order (del-6) filtering.  The blue curves use an even stronger, less scale-selective filtering (del-4).

6) In the second part of the paper, the authors try to quantify the $CO_2$ predictability in weather time scales and seasonal time scales. The discussions are lack of physical interpretation of the results and the implication of the results for flux inversion. Unlike meteorology fields and other air quality variables, such as O3, $CO_2$ transport and $CO_2$ observations are mainly used to quantify surface fluxes, not for prediction; the CO2 prediction itself does not have any applications. Under this context, it is important to illustrate the connection between $CO_2$ predictability discussed here and the $CO_2$ flux inversions.

**Response:**  We agree with the reviewer that it is important/necessary to identify the physical mechanisms which determine the $CO_2$ predictability.  Indeed, a focus of this work is on the processes connecting meteorological and $CO_2$ predictability. For this reason there is an attempt to address it in section 4 where we discuss the impact of convective mixing on the error spectra. Aside from the consideration of spatial scales, atmospheric processes from stratospheric sudden warmings, to tropical modes of predictability, to atmospheric waves have been invoked in the discussion of results.  It is exactly this type of analysis which is new to carbon cycle science and which is one of the novel features of this work.

A relevant matter for the flux inversion community is the use of transport model predictions (i.e. forecasts) in the computation of model-data mismatches. Transport model predictions are also performed with the fluxes retrieved from such an inversion to obtain $CO_2$ state estimates. The issue of transport model error (hence predictability) is of considerable concern to the flux inversion community. However, there may be a mismatch in terminology used by the weather prediction data assimilation community versus that used by the flux inversion community. To address this, a new section (section 2) was added to the revised manuscript to connect the concepts of predictability and transport error and to identify the components of transport error that are discussed in the article.

We recognize that the predictability of the $CO_2$ state has important implications for flux inversions. The new section connecting the concepts of predictability with transport error combined with the completely rewritten discussion section (the new section 6) address this issue. The abstract was also rewritten with this point in mind.

7) The authors simulate the analysis errors by shifting the met analysis by 6 hours. This would create large errors due to inaccurate description of diurnal cycle, which is especially significant over extratropics. Since ECCC has a hybrid approach to simulate background error covariance (page 10, line 24), the analysis error can be approximated by the spread of ensemble forecasts used in the hybrid scheme, which should be more realistic.

**Response:** The reviewer is correct that errors in the diurnal cycle would be created by shifting analyses by 6 h. We had considered this issue but decided it was not critical because the shifting is done everywhere at a given universal time, rather than local time. Thus the shift will be at different parts of the diurnal cycle for different longitudes, so the bias introduced by the shift averages out for the meteorological fields. It was assumed that even with the covariation of the wind fields with the $CO_2$ flux over 24 h, the impact when averaged over longitudes would again average out given that we only considered global diagnostics. However, the only way to know for sure if this is true, is to remove the diurnal cycle from the 6h analysis differences and rerun the perturbed analysis cycle. Therefore, we have done this. It is straightforward (though time consuming) to do this by simply removing the monthly mean of each hour from the perturbation. Figure R2-5 shows the impact of removing the diurnal cycle from 6h analysis differences on spatial spectra. The Figure corresponds to Figures 11-12 of the original manuscript with the red curves depicting the $CO_2$ spectra for differences in $CO_2$ due to 6h analysis differences, whereas the cyan curve depicts the impact from 6h analysis difference with the diurnal cycle removed. The month chosen is July 2009 because this is the month for which the impact of the diurnal cycle is the largest. In Fig. R2-5 (top row) there is an impact on the largest scales, with much reduced analysis uncertainty when the diurnal cycle is removed. The impact is largest near the surface where the boundary layer diurnal variation is important and in the stratosphere and above where tidal signals are important. However, the conclusion regarding the lack of predictability for smaller spatial scales (where the red or cyan curves cross the reference spectra (black or blue)) remains the same. Additionally, the conclusion regarding the lack of information at small spatial scales (the asymptoting of the red or cyan curves to the black one) remains the same. Because there is an impact from removing the diurnal cycle on predictable spatial scales, Figures 11-13 have been updated in the revised manuscript by replacing the red curve with the one corresponding to the new cycle with analysis perturbations having the diurnal cycle removed.

[Figure]

Figure R2-5: Spectra of $CO_2$ as a function of total (top row) or zonal (bottom row) wavenumber for July 2009 various model levels whose approximate pressure is given above each panel. The black curves represent predictability error while the red curves represent error due to a 6h shift in analysis. The red curves should be compared to the cyan curves which result when the diurnal cycle is removed from the 6h shift.

As stated in the original manuscript (p20, lines 9-11 and p25, lines 8-10), the ideal perturbation is a realization of analysis error, but dealing with operational systems poses significant logistic challenges. It is true that ECCC has a hybrid approach to obtaining background error covariances as of Nov. 18, 2014 but the hybrid system relies on a separate ensemble Kalman Filter system to provide realizations of forecast errors. It does not use nor does it compute analysis errors. However, ECCC also has a separate EnKF which is used for ensemble prediction along with the operational deterministic 4D-Var (i.e hybrid) analysis system. We had naturally considered the possibility of getting realizations of analysis error from the EnKF (though it uses lower resolution, a different model version and a lower lid from the deterministic system which produced the meterological analyses used here) but these were not archived by CMC in 2009-2010. The archives only start around 2011, and are not reliable so a full year of members could not be extracted. As noted in the original manuscript (p25, lines 11-16), we are devoting significant time right now to developing an ensemble Kalman Filter for greenhouse gas which, when ready, will give us exactly what we need. In the meantime, our proxy for analysis error (the 6h analysis difference minus the diurnal cycle) is useful (even though this proxy will be larger than analysis errors) since none of our main conclusions are affected by this choice, namely, that (1) some scales in $CO_2$ cannot be resolved due to the presence of analysis uncertainties, and (2) that for small enough spatial scales, the error spectrum due to analysis uncertainties asymptotes to the predictability spectrum hold regardless of the size of the error. Additionally, in the revised manuscript we stress that the proxy should over estimate analysis error.